# Nanofillers in Novel Food Packaging Systems and Their Toxicity Issues

**DOI:** 10.3390/foods13132014

**Published:** 2024-06-26

**Authors:** Xiangyu Zhou, Xiaoyu Zhou, Longli Zhou, Ming Jia, Ying Xiong

**Affiliations:** 1Department of Health Technology and Informatics, The Hong Kong Polytechnic University, Hung Hom, Hong Kong, China; 23069475r@connect.polyu.hk; 2The Fine Arts Academy, Hunan Normal University, Changsha 410012, China; rmhaxz5@ucl.ac.uk; 3Department of Metabolism, Digestion and Reproduction, Imperial College London, London SW7 2AZ, UK; lz121@ic.ac.uk; 4College of Computer and Mathematics, Central South University of Forestry and Technology, Changsha 410004, China; 5College of Food Science and Engineering, Central South University of Forestry and Technology, Changsha 410004, China

**Keywords:** nanofillers, improved food packaging, active food packaging, intelligent food packaging, degradable packaging, migration, toxicity

## Abstract

**Background**: Environmental concerns about petroleum-based plastic packaging materials and the growing demand for food have inspired researchers and the food industry to develop food packaging with better food preservation and biodegradability. Nanocomposites consisting of nanofillers, and synthetic/biopolymers can be applied to improve the physiochemical and antimicrobial properties and sustainability of food packaging. **Scope and approach**: This review summarized the recent advances in nanofiller and their applications in improved food packaging systems (e.g., nanoclay, carbon nanotubes), active food packaging (e.g., silver nanoparticles (Ag NPs), zinc oxide nanoparticles (ZnO NPs)), intelligent food packaging, and degradable packaging (e.g., titanium dioxide nanoparticles (e.g., TiO_2_ NPs)). Additionally, the migration processes and related assessment methods for nanofillers were considered, as well as the use of nanofillers to reduce migration. The potential cytotoxicity and ecotoxicity of nanofillers were also reviewed. **Key findings**: The incorporation of nanofillers may increase Young’s modulus (YM) while decreasing the elongation at break (EAB) (y = −1.55x + 1.38, R^2^ = 0.128, r = −0.358, *p* = 0.018) and decreasing the water vapor (WVP) and oxygen permeability (OP) (y = 0.30x − 0.57, R^2^ = 0.039, r = 0.197, *p* = 0.065). Meanwhile, the addition of metal-based NPs could also extend the shelf-life of food products by lowering lipid oxidation by an average of approx. 350.74% and weight loss by approx. 28.39% during the longest storage period, and significantly increasing antibacterial efficacy against *S. aureus* compared to the neat polymer films (*p* = 0.034). Moreover, the migration process of nanofillers may be negligible but still requires further research. Additionally, the ecotoxicity of nanofillers is unclear, as the final distribution of nanocomposites in the environment is unknown. ***Conclusions***: Nanotechnology helps to overcome the challenges associated with traditional packaging materials. Strong regulatory frameworks and safety standards are needed to ensure the appropriate use of nanocomposites. There is also a need to explore how to realize the economic and technical requirements for large-scale implementation of nanocomposite technologies.

## 1. Introduction

Food packaging is primarily designed to preserve and/or store food and protect it from physical, chemical, and biological contamination, thereby ensuring food quality throughout its life cycle, and maintaining its organoleptic properties [1,2]. Food packaging also serves as a container, facilitates transport, distribution, and warehousing, promotes marketing, and provides useful information and portion control to consumers [3].

The high prevalence of foodborne disease (i.e., nearly 600 million cases and 420,000 deaths each year worldwide [4]), global food crisis (i.e., almost 238 million people facing high levels of acute food insecurity [5]), and diversification of global food production has led to increasing demands for eco-friendly and safer food packaging. The main synthetic polymeric food packaging materials in use are easy to process, thermally flexible, lightweight, cost-effective, and can be chemically processed into a variety of lengths and shapes with functions such as antioxidants to meet the varying needs of food products [6]. For example, the functionalization of polyvinyl chloride (PVC) polymer with additional bioactive natural antioxidants vanillic acid, cinnamic acid, coumaric acid, caffeic acid, and naringenin prolonged the oxidation of linseed oil by 1 to 6 days over virgin PVC [7]. Polypropylene (PP) grafted with tannic acid showed similar antioxidant effects and slowed down the oligomerization of linseed oil [8]. Hazer and Ashby [9] also reported that menthol and lipoic acid-modified PP and PVC (i.e., PP-Mnt, PP-Lip, PVC-Mnt, and PVC-lip) could be applied as antioxidant packaging materials for vegetable oils and oil-containing foods. However, some synthetic polymer-based plastic packaging (e.g., polyester: polyethylene terephthalate (PET), polyethylene (PE)) have inherent defects in poor mechanical properties, cytotoxicity and ecotoxicity risk, slow degradation, and other environmental concerns. Thus, it is imperative to develop non-toxic, recyclable, sustainable, biodegradable, and more environmentally friendly alternatives, such as biopolymers, for food packaging.

Biopolymers offer the potential for more environmentally friendly food packaging materials due to their better biodegradability, non-toxicity, and sustainability compared to plastics [6]. They are directly extracted from polysaccharides (e.g., cellulose, starch, alginate, and chitosan) and proteins (e.g., casein, gluten, and gelatin), synthesized from renewable biomass-derived monomers (e.g., poly (lactic acid) (PLA)), and produced by fermented products of natural or genetically modified microorganisms (i.e., polyhydroxyalkanoates) [6]. They can control moisture and gas exchange, retain aromas and texture, and have antimicrobial activity [10]. Nevertheless, in most applications, these biopolymers lose these properties in high-humidity environments due to their hydrophilic nature of the structural groups. Meanwhile, their mechanical properties are usually barely comparable to the strength of the plastic materials currently applied in industry [11]. Therefore, recent research has concentrated on formulating biocomposites (via blending or laminating), using the different properties of the materials (e.g., polymer blends, particles, cross-linking agents, and plasticizers) to compensate the limitations of each technology usually used in isolation [11]. For example, Jiang et al. have crosslinked poly(vinyl alcohol) (PVA), citric acid, and chitosan into a composite film. It had better permeability (WVP: 5.26 × 10^−4^ g h^−1^ m^−2^ Pa^−1^ mm), mechanical properties (tensile strength (TS): approx. 16.5 MPa), and creep resistance (EAB: approx. 570%) than the PVA-free film (WVP: 8.08 × 10^−4^ g h^−1^ m^−2^ Pa^−1^ mm, TS: approx. 7.5 MPa, EAB: approx. 500%) [12]. This film prolongs the shelf-life of cherries (up to 13 days) compared to unpackaged cherries, which become dehydrated (wrinkled) and discolored after 7 days [12].

Additionally, the addition of discontinuous nanofillers (e.g., NPs, nanocrystals, nanofibers, nanotubes) to continuous biopolymers could generate bio-nanocomposites with at least one external size ranging from 1 to 100 nm. They are structural materials with higher homogeneity and structural interactions, providing better mechanical, barrier, and antimicrobial properties than the original biopolymers [3]. The high aspect ratio (L/D) of nanofillers also increased the interfacial area between the nanofillers and polymer matrix, which increased load transfer and stress distribution in the nanocomposites, leading to increased stiffness, strength, and toughness [13]. For example, the addition of 1.5 wt% MMT to 6% *w*/*v* corn starch and 2% *v*/*v* lemongrass oil nanoemulsions significantly increased the TS and EAB of bio-nanocomposite films by 37% and 16.50%, respectively, as compared to the control [14]. The high surface area-to-weight ratio of nanofillers enhances the interaction between the nanofillers, polymer chains, gases, and liquids, thereby improving the barrier properties [3]. For example, cellulose nanocrystal (CNC)-coated paper with immobilized AgNPs reduced WVP and OP by about 45% and 93%, respectively, and extended the shelf-life of strawberries by up to 7 days compared to unpackaged strawberries [15]. Furthermore, the large surface area of nanofillers promotes their accumulation on the microbial cell surface. Meanwhile, their nano-size enables them to penetrate the cell membranes more easily, interact with microbial proteins and enzymes, disrupt their normal functions, generate reactive oxygen species (ROS), and inhibit microbial growth [16]. For example, pectin/AgNPs nanocomposite films significantly inhibited *Escherichia coli (E. coli)* (diameter of inhibition zone (DIZ) (mm): 8.4 ± 1.2 mm) and *Saccharomyces cerevisiae (S. cerevisiae)* (DIZ: 3.9 ± 0.8 mm) [17]. Additionally, the addition of natural pigments (e.g., anthocyanins) to the nanocomposite film can change the color with pH and ammonia concentrations, thus indicating food freshness [18].

However, nanofillers may migrate from food packaging into food products because their decreased size and increased surface area per unit volume result in shorter diffusion distances (x) and higher concentration gradients ($C_{p}$) according to Fick’s second law (Equation (1)). This raises concerns about consumer health and safety [19]. Equation (1) describes the non-stationary diffusion of substances, i.e., changes in substance concentrations over time and space [20].
(1)$$\frac{\partial C_{p}}{\partial t}=D(\frac{\partial^{2}C_{p}}{\partial x^{2}})$$
C_P_: the concentration of the migrating material in the polymeric material; D: the diffusion coefficient; t: the penetration duration; x: the distance between the package and the food. 

Furthermore, Bott, Störmer, and Franz [21] further reported that the NPs migration process mainly depends on the partition coefficient (k_p_) between the polymer and the food and their D_p_ in the polymer, which is determined by the temperature and molecular size of the NPs (Equation (2)). Meanwhile, their experimental findings and migration modeling showed that NPs are unlikely to migrate from food contact plastics [22]. This is consistent with the findings of a systematic review [23] but different from some studies in a review [24].
(2)DP,i=Duexp⁡(wi,e−wp,e·0.1414j+223−wwj,e23Tm,pRRT
i: an abbreviation for substance i (i.e., NPs in this study); w_i,e_ = (1 + 2π/i)^i/e^, j = (i^1/3^); w_j,e_ = (1 + 2π/j)^j/e^; p = (M_r,p_/14)^1/3^; w_p,e_ = (1 + 2π/p)^p/e^; i = (M_r,i_ − 2)/14; w = e^2π/e^ and D_u_ = 1 m^2^ s^−1^; M_r,i_: relative molecular mass of the migrating substance (i); M_r,p_: relative molecular mass of the polymer (p); T_m,p_: melting temperature of the polymer; R: gas constant; T: absolute temperature in Kelvin; 0.14 (14j + 2)^2/3^: relative molar cross-sectional of the diffusing particles.

Therefore, various nanofillers can enhance the mechanical and barrier properties of nanocomposite films, enhance their antimicrobial properties, and help to display the freshness of packaged foods with less environmental impact. However, the potential migration of the nanomaterials into the food and the associated cytotoxicity requires further research. It is necessary to summarize the latest research results to keep abreast of the latest developments in nanocomposites for novel food packaging (Graphical Abstract). 

## 2. Use of Nanofillers for Improvement of Food Packaging

Nanocomposites have higher mechanical strength, gas barrier properties (e.g., O_2_, CO_2_, volatiles, and flavor) and UV barrier properties compared to original biopolymers [19,25] (Figure 1), and commonly used nanofillers include nanoclays, carbon nanotubes, cellulose-based nanofibers or nanowhiskers, starch nanocrystals, chitosan NPs, and silicon dioxide NPs (SiO_2_ NPs). 

The mechanical properties of nanocomposites mainly consist of TS, YM, and EAB, which indicate the maximum mechanical stress, the resistance to linear or uniaxial elastic deformation, and the deformation range of the packaging film, respectively [26]. For nanocomposites made with specific biopolymers, TS, YM, and EB show the extent of their rigidity and flexibility. Greater TS and YM show stronger molecular linkages and less fluidity, while larger EB shows looser linkages and greater fluidity [26]. Meanwhile, the addition of different NPs can improve the rigidity of the nanocomposites by enhancing the interaction force between the film matrices, resulting in increased TS and YM and decreased EAB [26].

Gas barrier properties include water vapor and oxygen barrier properties, which are primarily associated with the curvature degree of the pathway diameters in the polymer network structure, calculated by the Nielsen equation (Equation (3)) [27]. Maintaining normal moisture levels in fresh foods is vital to avoid quality deterioration caused by water loss, while preventing some dried foods (e.g., bread) from absorbing water and oxygen from the environment helps to prevent oxidative deterioration of packaged food [26].
(3)$$\frac{P_{c}}{P_{m}}=\frac{1-V_{f}}{1+\frac{L}{2D}V_{f}}$$

P_c_: the permeability of the polymer composite, P_m_: permeability of the unfilled polymer matrix, V_f_: the volume fraction of filler, $\frac{L}{D}$: the aspect ratio (length/thickness) of the filler.

Determining the transmittance of the nanocomposite films at selected wavelengths from 200 to 800 nm allows to investigate their UV–visible light barrier properties. This is because the high light transmittance of nanocomposites in the visible region directly reflects consumer judgment of the visual appearance of fresh produce. Next, the low UV transmittance of nanocomposites can reduce the oxidization and decomposition of foodstuffs due to UV irradiation [26].

### 2.1. Nanoclay

Nanoclay was one of the first nanocomposite polymers to enter the market. For example, Nanocor Inc. (Chicago, CA, USA) used nanoclay in multilayer PET bottles and sheets for food and beverage packaging to minimize the CO_2_ loss from the drink and the O_2_ ingress into the bottles [28]. Currently, nanoclay is also one of the most commonly used NPs in food packaging, but it works toward balancing the high cost and improved mechanical properties of food packaging [28]. 

The natural clay material called montmorillonite (MMT) is widely applied as a nanofiller in food packaging due to its large specific surface area and large L/D (50–1000) [29]. MMT consists of an octahedral layer of aluminum hydroxide or magnesium hydroxide located on a shared edge between two silicon oxide tetrahedra [25,30]. The clay structure of MMT is formed by the periodic accumulation of hundreds of negatively charged silicate lamellae into particles or tentacles with a diameter of 8–10 mm [31]. The imbalance of the negative surface charge is due to the isomeric substitution of Si^4+^ for Al^3+^ or Al^3+^ for Mg^2+^ within the silicate layer, which is compensated by exchangeable cations (usually Na^+^ and Ca^2+^) that occupy the interstitial layer space [32]. This high surface-to-volume ratio and cation-exchange properties give MMT suitable miscibility and load-bearing capacity with cationic polymers [32]. MMT is also often chemically modified to induce swelling of the clay, which improves the dispersion of MMT in the polymer matrix and allows better intercalation and potential exfoliation of the polymer chains into the expanded clay structure [33,34,35]. Delamination of the MMT silicate layer can be composited with polymers to form nanocomposite films, while their mechanical properties are also enhanced by strain alignment of MMT with polymer chains [36]. For example, the addition of 2 wt% of Cloisite 15A nanoclay to *Salvia miltiorrhiza* seed mucilage nanocomposite film increased YM, EAB, and TS by about 20%, 42%, and 58%, respectively, as compared to composites without NPs.

However, the optimal concentration and type of nanoclay remain to be determined. For example, the addition of 3 wt% of Cloisite 15A nanoclay decreased the EAB and TS of nanocomposite by 31% and 7%, respectively, as compared to the addition of 2 wt% [37]. Consistently, the addition of 1 wt% of halloysite to the xylan–arginine matrix also increased the YM by ~10.87% and TS by ~24.97% compared to the control film. The incorporation of 3 wt% and 5 wt% of halloysite decreased the YM and TS values compared to the 1 wt% addition of halloysite [38]. Nevertheless, the same study reported that the incorporation of Bentonite into the nanocomposite films led to a linear increase in TS and YM values. The optimum concentration of 5 wt% increased the TS and YM by 112.62% and 76.25%, respectively, in comparison to the control film [38]. 

Additionally, the presence of van der Waals gaps (~1.26 nm) between the stacks formed by multiple clay layers can create a tortuous diffusion channel, which greatly reduces the permeation of gases, moisture, flavors, and other small molecules through the food packaging material, ultimately extending the shelf-life and improving the quality of the packaged food [30,39]. Gaume et al. [40] showed that the incorporation of 5 wt% MMT-Na+ (Cloisite^®^ Na+) MMT to PVA reduced OP and WVP by almost 3 and 6 times, respectively, compared to virgin PVA. Chandio et al. [41] further applied MMT-Na+ to water-soluble poly(vinyl alcohol) (PVOH), where the use of 10 wt% MMT-Na+ reduced the WVP by a factor of about 7 and was economical. However, the addition of 2.5 wt% MMT-Na+ only reduced the WVP and OP of the pectin nanocomposite films by approximately 33% and 26.85% (i.e., a factor of 1.49 and 1.37) compared to the neat polymer, respectively [42]. Similarly, the addition of 6–8 wt% MMT to pomegranate peel pectin reduced WVP by 30–40% (i.e., a factor of 1.43–1.67) [43]. These findings suggest that the polymer matrix, NPs loading, and polymer–NPs interactions should be considered when improving the final permeation/barrier properties of nanocomposites.

### 2.2. Carbon Nanotubes

Carbon nanotubes (CNTs) are cylindrical nanomaterials rolled up from carbon allotropes with diverse nanoscale diameters (2–80 nm). The number of concentric tubes determines the different types of CNTs, i.e., one-atom-thick single-walled carbon nanotubes (SWCNTs) or multi-walled carbon nanotubes (MWCNTs) [44]. The incorporation of CNTs as nanofillers in polymers improved the mechanical, thermal, and conductive properties of nanocomposites due to their exceptional TS up to 500 GPa, YM up to 1 TPa, and thermal conductivity of 3500 W m^−1^ K^−1^ [45]. For example, the addition of 1 wt% CNT significantly improved the TS and YM properties of polylactic acid and poly (ε-caprolactone) packaging films by about 25.54% and 28.41%, respectively [46]. Consistently, the addition of 0.3 wt% ZnO-doped multi-wall nanotubes (MWCNTs-ZnO) significantly increased the TS of the PVA nanocomposite film by 116% compared to that of the pure PVA film [47]. Furthermore, the EAB of the PVA film modified with 0.9 wt% MWCNTs-ZnO was also significantly increased by 81% compared to that of pure PVA films since MWCNTs-ZnO helps to stretch the chain or improve the orientation of the polymer. Meanwhile, the interaction of MWCNTs-ZnO with hydrogen bonding in PVA may form interchain bonding, which enhances the cohesion of the PVA network and strengthens the nanocomposites, thus requiring a greater tensile force to pull it off [47]. The 0.6 wt% of MWCNTs-ZnO could also reduce the WVP of PVA nanocomposite film, making it easier to retain the moisture of fresh vegetables for more than 4 days compared to pure PVA films. This is because nanofillers increase the tortuous path for water molecules through the nanocomposite film [47].

Additionally, CNTs were discovered in 1991. However, the potential environmental and health risks, as well as the benefits of using CNTs in food packaging, have not been fully assessed within the context of the traditional Life Cycle Assessment (LCA) combined with Risk Assessment (RA) [48]. The potential cytotoxicity of CNTs to human cells (at least in contact with the skin) due to the possibility of unintentional migration has been a concern, limiting their commercial application in food packaging [49]. The release of carbon NPs from nanocomposite film has been extensively assessed over the past decade under different time, temperature conditions, and food simulants [50,51]. 

Surface functionalization and modification of CNTs can improve their compatibility and interaction with the polymer matrix, making them biocompatible and non-toxic to a certain extent [52]. Specifically, techniques such as polymer grafting/encapsulation, polynuclear aromatic compounds, surfactant adsorption, and NPs decarbonization can modify van der Waals forces and π-stacking interactions, thereby altering the surface properties of CNTs and improving their compatibility in the polymer matrix [52]. Meanwhile, the carboxyl or amino modification of CNTs has been reported to help PLA-based materials change the stiffness storage modulus and mechanical properties [53]. In addition, the presence of metal impurities in CNTs may lead to differences in cytotoxicity, thus requiring special consideration for the development of improved purification techniques such as neutron activation analysis, ICP-MS, etc. [52]. 

### 2.3. Cellulose-Based Nanofibers or Nanowhiskers

Since the first announcement of using cellulose whiskers as a reinforcing phase by Favier et al. [54]. in 1995, the use of cellulose-based nanowhiskers has yielded novel nanocomposites with enhanced properties and has triggered research into starch nanocrystals and chitosan NPs (CNPs) [55]. Cellulose-based nanomaterials are widely found in plant cell walls and are also ubiquitous and strong natural polymers with unique characteristics, such as being cost-effective, environment-friendly, and easy to recycle [56,57]. Meanwhile, the nanoscale structure and the high specific surface area of cellulose give the cellulose nanocomposites superior mechanical and barrier properties, even at low cellulose nanofiller loadings. The agglomeration problem caused by high nanofiller loadings can also be avoided [58]. Therefore, although there were relatively few reports on these cellulose-based nanomaterials in the literature prior to 2010 compared to nanoclays, the last decade has seen a much more rapid growth in their use and research [55]. 

The nanocellulose isolated from cellulose can be divided into cellulose nanofibers and cellulose nanowhiskers, depending on the chemical and mechanical production process [59]. Cellulose nanofibers are prepared by mechanical shearing, which disintegrates the hydrogen bonds between the cellulose chains, leading to further aggregation of the elementary fibrils to form the nanofibers containing amorphous and crystalline regions [60]. On the other hand, cellulose nanowhiskers are very small crystalline nanoparticles made by a chemically based process of acidic or oxidative hydrolysis, whereby the amorphous regions of cellulosic materials can be rapidly hydrolyzed to form isolated and pristine crystalline regions [61]. 

Sucinda et al. [62] reported that adding 1 wt% Napier cellulose nanowhisker (NWC) resulted in TS and YM of the film being approximately 5.78% and 7.48% higher than those of pure PLA films. Meanwhile, PLA/1.0 wt% NWC bio-nanocomposite films showed the lowest UVA and UVB transmittance of 7.44% and 3.65%, respectively, without affecting the transparency (λ_800_ = 20.32%). This is because 1 wt% of NWC is well dispersed in the PLA matrix and therefore does not lead to aggregation affecting transmission. It increases nanofiller interaction with the matrix, providing more free space between the PLA polymer chains, which improves chain mobility and enhances mechanical properties. Consistently, Haghighi et al. [63] showed that the addition of 7.5 wt% bacterial cellulose nanowhisker (BCNW) improved the TS, EAB, and YM of gelatin/PVA blend-based bio-nanocomposite films by about 21.56%, 41.18%, and 19.10%, respectively, compared with that of pure gelatin/PVA blend films. However, the addition of 10wt% BCNW resulted in a significant decrease in the mechanical properties of the composite nanofilms due to the disruption of the cross-sectional morphology caused by the BCNW agglomeration [63]. Furthermore, Li et al. [64] developed a novel cellulose nanofibers and corn straw core nanocomposites (CNF/CSC)-100 (i.e., 100 mg CSC content in nanocomposites), which increased TS by 1.25-fold, 4.71-fold, and 8.04-fold compared to PP, poly (butylene adipate-co-terephthalate) (PBAT), or poly (butanediol sebacate—butanediol terephthalate) (PBSeT), respectively. Meanwhile, the WVP properties of CNF/CSC-100 film were also improved by 46.59% over PBAT.

### 2.4. Starch Nanocrystals

Prior to 2010, there were far fewer studies on starch nanocrystals than on cellulose-based nanomaterials [55], but in the last 10–15 years, the research and development related to starch nanocrystals for food packaging applications has grown rapidly.

Starch is a relatively pure, non-toxic, and inexpensive raw material that does not require the same rigorous purification as lignocellulosic materials [65]. However, the isolation of starch nanocrystals requires a longer acid hydrolysis process compared to cellulose, as the crystalline lamellae of starch are more resistant to acid hydrolysis compared to the amorphous regions. Therefore, optimizing the decomposition process and transitioning from the micron to the nanoscale is an area for further research. Meanwhile, these starch crystalline particles, which have been described as the remnants of nanocrystals, contain a platelet-like morphology with a length of 20–40 nm, a width of 15–30 nm, and a thickness of 5.7 nm [66]. This platelet-like morphology can provide other beneficial properties, e.g., improved barrier properties. Additionally, since most of the work prior to 2010 was conducted using waxy maize/corn starch, there is an opportunity to explore other sources of starch in food packaging to improve food preservation. For example, according to methods provided by Zhou et al. [67], the 0.1 wt% rice starch nanocrystals were prepared through acid hydrolysis (H_2_SO_4_) for 7 days, which were incorporated into native starch films, resulting in an increase in TS and EAB of 26.8% and 991.5%, respectively [68]. This is because starch nanocrystals can be used as structural enhancers to form three-dimensional (3D) networks with cross-links in the film structure, thereby increasing fracture re-resistance and improving the elongation of the material [68]. Furthermore, the WVP of 0.1 wt% rice starch nanocrystals was 64% lower than the control because the addition of a low concentration of rice starch nanocrystals changed the structure of the film and reduced the penetration of water molecules through the film structure [68]. 

Consistently, Gharibzahedi et al. [69] fabricated spherical starch nanocrystals (SNCs) from potato starch granules (PSGs) by ultrasound-assisted acid hydrolysis and developed new nanocomposite films by incorporating them to whey protein isolate (WPI) and purified jujube polysaccharide (JPS). They reported that the best mechanical properties (i.e., TS: 14.0 MPa, EAB: 14.87%, and YM: 47.80 MPa) and suitable WVP (0.63 × 10^−10^ g/m Pa) and OP (9.62 cm^3^/m^2^ d atm) were obtained using 10.0% WPI, 12.72% JPS, and 1.31% SNC, resulting in better retention of physicochemical, sensory, and microbiological properties of freshly cut carrots during cold storage up to 14 days. However, the addition of higher concentrations (e.g., 12 wt%) of starch nanocrystals during film formation leads to an increase in nanocrystal agglomeration, which may result in the formation of larger pores in the film and promote gas passage through the film. As a result, the WVP value of the film cannot be significantly increased (12 wt%: 4.2 ± 0.2 10^−12^·g·m·m^−2^·s^−1^·Pa^−1^ vs. 10 wt% 4.2 ± 0.1 10^−12^·g·m·m^−2^·s^−1^·Pa^−1^) [70].

### 2.5. Chitosan NPs

Chitosan is an N-deacetylated derivative of chitin, which is mainly derived from seafood by-products such as crab and shrimp shells and fungal cell walls [71]. Chitosan has more functional groups, such as chelated amino groups, which contribute to its antimicrobial and mechanical properties. Therefore, chitosan is a commercially available, inexpensive, non-toxic, and biodegradable biopolymer used in food packaging [72]. CNPs are basically fabricated by modifying chitosan biopolymers via tripolyphosphate (TPP) cross-linking, microemulsion/reverse micelle technique, precipitation/coacervation methods, and emulsification approach [73]. These prepared CNPs have excellent mechanical, barrier, and thermal properties, mainly due to their improved surface area, which increases the interactions between CNPs and polymer matrix [71]. However, similar to starch nanocrystals, there were few studies on the incorporation of CNPs into food packaging before around 2014 [74,75]. However, the number of relevant studies has increased over the last decade. For example, the 5 wt% CNPs (chitosan: TPP mass ratio = 3:1) could be applied as a physical cross-linker to enhance the entanglement of the polymer chains in the hydroxypropyl methylcellulose and hydroxypropyl starch (HPMC/HPS) matrices, which significantly increased the TS and EAB by 39.46% and 229.16%, respectively, compared to the control [76,77]. Meanwhile, the WVP value of 5 wt% CNPs/HPMC/HPS films was reduced by about 33% compared to the control, which was attributed to the formation of hydrogen bonds between chitosan and the HPMC film matrix, or the smaller-sized NPs were better able to occupy the voids in the porous HPMC film matrix [76]. Consistently, Roy et al. [78] showed that the incorporation of 1 wt% of curcumin-integrated CNP(CNP@Cur) significantly increased the TS, EAB, and YM of pullulan/chitosan-based functional composite film by 15%, 7%, and 4%, respectively, while the WVP was reduced by 35%.

Furthermore, CNPs are formed by ionic gelation, where the positively charged amino groups of chitosan electrostatically interact with the polyanions that act as cross-linkers [79]. Therefore, this polycationic characteristic of chitosan allowed the chitosan NPs to exert broad antibacterial activities, including *L. monocytogenes*, *E. coli*, *S. aureus*, *P. microbilis*, and *P. vulgaris*, etc. [78,80,81].

### 2.6. Silicon Dioxide NPs (SiO_2_ NPs)

Silicon is commonly found in soil deposits on the earth’s surface, normally in the form of SiO_2_ in nature [82]. Due to their mesoporous nature, high specific surface area, and low toxicity, SiO_2_ NPs were investigated as inorganic nanofillers to improve the mechanical and barrier properties of nanocomposite films during the first decade of their initiation [83,84,85]. Starting around the 2010s, its food packaging applications expanded to be incorporated into the bio-based polymeric matrix, etc. 

For example, due to the strong hydrogen interaction between SiO_2_ NPs and the polymer matrix, Hou et al. [86] reported that the TS of agar/sodium alginate (SA) nanocomposite film increased from 45.18 MPa to 74 MPa with the increase of SiO_2_ NPs addition concentration from 0 to 10 wt%. Meanwhile, SiO_2_ can be evenly distributed in the molecular chain of the film, explaining the gradual increase in the EAB of the film from 33.04% to 52.99%. Consistently, Júnior et al. [87] also reported that the incorporation of 10 wt% SiO_2_ to SA/hydrolyzed collagen (HC) blend films significantly increased the TS and EAB by 39.56% and 83.59%, respectively, compared to the control. However, high concentrations (e.g., >10 wt%) of SiO_2_ can result in silica aggregation in the film, forming stress concentration points and disrupting the functional structural integrity of the film matrix network. This can reduce the degree of cross-linking between components of the polymer matrix, thereby reducing TS and EAB, i.e., 109.67 ±  7.89 MPa and 10.28  ±  0.28% for SA/9 wt% SiO_2_ NPs, while 87.20  ±  1.77 MPa and 5.60  ±  0.57% for SA/18 wt% SiO_2_ NPs [88]. Moreover, the addition of 1–3 wt% SiO_2_ to soy protein isolate-based films progressively decreased the WVP and OP of the films [89]. Since NPs can occupy the pores of the film matrix and form a dense network structure, the path of water molecules in the film matrix can be extended, thus reducing the diffusion of water molecules [90]. Meanwhile, the small-sized NPs are more easily and uniformly dispersed into the macromolecular chains, which is beneficial to improving the barrier property of the film. Therefore, the size (average diameter) of the hybrid nanomaterials (SiO_2_–gallic acid NPs) decreased from 408.7 ± 3.20 nm to 112.7 ± 0.55 nm at the same concentration (8 mg/mL), leading to a decrease in the WVP of chitosan films from 4.04 × 10^−8^ g·m/h·m^2·^Pa to the 3.82 × 10^−8^ g·m/h·m^2^·Pa [91]. 

The relationship between the changes in YM and EAB, the changes in WVP and OP when adding nanofillers into nanocomposite film can vary because of the type of nanofillers, their concentration, their dispersion degree in the polymer matrix, and the film processing conditions. Table 1 shows more relevant studies indicating that in many cases, the incorporation of nanofillers increases YM while decreasing EAB (r = −0.358, *p* = 0.018, Figure 2a) and decreasing the WVP and OP (r = 0.197, *p* = 0.065, Figure 2b), as compared to the virgin nanocomposite films. However, the weak correlations (i.e., r < 0.5) suggest that further studies are needed to confirm the findings.

## 3. Use of Nanofillers in Active Food Packaging

Active packaging is designed to extend shelf-life and maintain the quality and safety of fresh food. There are three types: (1) scavenging systems produce the desired reaction in the food system without migration of the active ingredient from the package into the food product, e.g., by absorbing oxygen, ethylene, and water; (2) releasing systems control the migration of non-volatile compounds or the emission of volatile compounds into the environment around foodstuffs, e.g., CO_2_ emitters and ethylene emitters; and (3) antimicrobial systems [109]. 

Antimicrobial packaging is one of the most promising applications of active packaging technology, which is generally synthesized majorly by incorporating NPs derived from either metal or metal oxide, as well as essential oil (EO) loaded nanoemulsions [109]. 

Several antimicrobial mechanisms have been proposed for different nanometals and have been studied based on the morphological and structural changes in bacterial cells (Figure 3). Firstly, NPs can anchor and penetrate the bacterial cell wall, causing structural changes in cell membrane permeability and leading to cell death [110]. Some NPs can also be adsorbed onto bacterial surfaces, releasing positively charged metal ions that react with bacterial cell membranes with negative charges, disrupting the integrity of the bacterial outer and cytoplasmic membranes and generating ROS [110]. Secondly, the ROS on the surface of NPs generate oxidative stress, which disrupts cell membranes, making them porous, inhibiting cell wall synthesis, and promoting protein and DNA damage [111]. Thirdly, metal NPs enter cells and release metal ions, which can interact with sulfur- and phosphorus-containing compounds such as DNA bases in bacterial cells, resulting in DNA destruction and ROS generation [112]. Next, metal NPs could dephosphorylate the peptide substrates on tyrosine residues, thereby inhibiting signal transduction [110]. Finally, metal NPs may also exert their antimicrobial activity by decreasing ATP levels through the collapse of membrane potential and electron transport, suppression of ATPase activity, and inhibition of ribosomal subunit binding to tRNA [113].

### 3.1. Silver NPs (Ag NPs)

Ag NPs have a large surface area, especially triangular and spherical Ag NPs. They can accumulate considerably in the bacterial membrane, releasing Ag+, which is taken up by the bacterial cell. The intracellular Ag+ concentrations varied over time, reaching a maximum at 36 h (approximately 12.4 times the extracellular Ag+ concentration), while the extracellular Ag+ concentrations gradually decreased [114]. The addition of AgNPs to a composite mixed with hesperidin and pectin (3:1) released approximately 4 ppm Ag+ within 6 h, leading to increased intracellular ROS in *E. coli*; and 6 ppm Ag+ within 24 h, leading to protein leakage of *E. coli* up to 27.7 μg/mL [114]. Hong et al. [115] also reported that the number of viable *E. coli* cells in beef samples wrapped with 2 wt% Ag NPs/agar films was significantly reduced by approximately 95% compared to those wrapped with pure agar films throughout the storage period (15 days at 5 °C). Meanwhile, Grisoli et al. [116] reported that the release of 23.9% Ag+ (of the total dose) (i.e., ~7.57 ppm) from the 1 wt% AgNPs film explained its greater antimicrobial efficacy against both *E. coli* and *S. aureus* (microbial effect: ME > 7) compared to blank films (ME: 2.3 for *E. coli*; ME: 3.2 for *S. aureus*).

Furthermore, in contrast to Gram-positive bacteria, Gram-negative bacteria are more susceptible to Ag NPs due to their thinner layer of peptidoglycan surrounded by an outer membrane containing lipopolysaccharides (LPS), which can promote the intreatcion and penetration of Ag NPs [117]. For example, Huang et al. [118] showed that gliadin@AgNPs continuously released Ag+ over a 24 h period, with a cumulative release of up to 16.7% of the total release (i.e., 1.25 ppm) and showed higher antimicrobial activity against Gram-negative *E. coli* than against Gram-positive resistant *S. aureus*. Similarly, surface-immobilized Ag NPs of the same dimension with Grisoli et al. [116] released 15.1% Ag+ over 24 h with ME values of 5.90 and 5.54 for *E. coli* and *S. aureus*, respectively [119]. Raghav et al. [117] also reported that AgNPs–carboxyl cellulose nanofibers (AgNPs-CCNFs) hybrid materials exhibited greater antimicrobial activity against *E. coli*. (zone of inhibition(ZOI): 103.87 mm^2^) than Gram-positive *E. faecali* (ZOI: 65.04 mm^2^).

In addition, the antimicrobial properties of Ag+ can be improved by four orders of magnitude by incorporating small, uniformly dispersed silver nanofillers on MMT nanoclay compared to in situ precipitated bare silver particles, allowing precise control release of nano-antimicrobial Ag+ [120]. 

### 3.2. Zinc Oxide NPs (ZnO NPs)

ZnO NPs show advantages over Ag NPs for food packages due to their low toxicity to humans (GARS material (generally recognized as safe)) and relatively low cost [121]. Similar to Ag NPs, their antimicrobial effect is derived from the release of Zn^2+^ ions. Their cytotoxicity is also dependent on the interfacial potential. Since the teichoic acid of Gram-positive bacteria is attached to the peptidoglycan or the underlying plasma membrane, their cell membranes are more negatively charged than those of Gram-negative bacteria. The ZnO NPs generate more ROS when internalized into Gram-positive cells, leading to cell wall disruption and cell membrane damage [122]. For example, the addition of 12.5 wt% ZnO NPs to gelatin/starch-based film significantly increased the ZOI against both *E. coli* (85.30 ± 18.90 mm^2^) and *S. aureus* (67.28 ± 17.28 mm^2^) compared to the control (both: 0 ± 0 mm^2^) [123]. Similarly, Hashem et al. [124] also showed an antimicrobial effect against both *E. coli* (DIZ: 15.1 ± 0.76 mm) and *S. aureus* (DIZ: 12.1 ± 0.71 mm) after incorporating 4 mL of ZnO NPs into HPS/PVA/PA films. Furthermore, the smaller sized ZnO NPs have a greater surface area and chemical reactivity. They enhanced the release of Zn^2+^ by about 272% at 24 h and were more effective against *S. aureus* than larger sized ZnO, i.e., ZnO NPs with a DIZ of 19.67 ± 0.58 mm vs. bulk ZnO’s 10.00 ± 1.00 mm, respectively, at a concentration of 5000 µg/mL [125]. 

### 3.3. Other NPs 

Alongside Ag and ZnO, TiO_2_ NPs are non-toxic to humans (GARS) [126]. TiO_2_ NPs are one of the most common photocatalysts with stable and low-energy valence band electrons and an empty conduction band. The band gap energy represents the energy difference between the valence and conduction bands, i.e., the minimum light energy required for a redox reaction to occur at the TiO_2_ surface during photocatalysis [127]. When TiO_2_ absorbs UV light with energies greater than the band gap energy, electron-hole pairs are generated, which react with O_2_ or H_2_O adsorbed on the TiO_2_ surface to form ROS, such as superoxide radicals and hydroxyl radicals [128]. These ROS then oxidize the unsaturated polyphospholipid component of microbial cell membranes and DNA, ultimately causing damage to bacterial cells [128]. For example, the addition of 1 wt% of TiO_2_ NPs to the chitosan film prepared with *Cymbopogon citratus* essential oil (1.5% *v*/*v*) helped to reduce the total number of bacteria by around 1 log cycle compared to the control after 10 days of storage [129]. The cellulose nanofiber/whey protein matrix containing 1 wt% TiO_2_ and 2% w/v rosemary oil could also significantly reduce lipid oxidation and lipolysis of the lamb meat during storage, extending its shelf-life from around 6 days (control) to 15 days [130].

Copper oxide nanoparticles (CuO NPs) are FDA-approved nanoparticles with excellent antimicrobial properties because of their high specific surface area and the release of copper ions [131]. The embedded 5 mM CuO NPs in 3 wt% of sodium alginate (SA) and 0.5 wt% of CNW showed a high inhibitory effect against *S. aureus* (DIZ: 27.49 ± 0.91 mm), *E. coli* (DIZ: 12.12 ± 0.58 mm), *Salmonella* sp. (DIZ: 25.21 ± 1.05 mm), *C. albicans* (DIZ: 23.35 ± 0.45 mm) and *Trichoderma* spp. (5.31 ± 1.16 mm), thus preventing microbial contamination in freshly cut peppers [132]. Meanwhile, the addition of Ag, ZnO, and 2 wt% CuO NPs to the starch polymer matrix produces a synergistic antimicrobial effect, inhibiting the growth of *E. coli* and *S. aureus* more efficiently (i.e., a reduction of 91–94%) compared to the same wt% of the individual NPs (i.e., a reduction of 83–87%) [133]. 

Gold nanoparticles (Au NPs), also known as gold colloids, range in size from 1 nm to 100 nm [134]. Chowdhury et al. [135] reported that Au NPs enhanced the antimicrobial activity of PVA–glyoxal films against *E. coli* with DIZ of 13 mm, boosting the shelf-life of bananas up to 5 days. In the PVA film (FilmMIX-Ag/GNS) containing Ag NPs and gold nanostars (GNS), the release of gold was negligible, but the release of Ag+ was slow and persistent, resulting in a synergistic antimicrobial effect against *E. coli* and *S. aureus*, with an ME value of more than 7 over 24 h [116].

The crystal morphology of iron oxide (Fe_3_O_4_) NPs includes a large number of edges, corners, and potential reactive sites, which can contribute to their antimicrobial properties [136]. Fe_3_O_4_ NPs can interact with cell membranes and penetrate into the cell interior, leading to membrane damage and bacterial inactivation [137]. El-Khawaga et al. [137] modified Fe_3_O_4_ NPs with chitosan and the synthesized chitosan–Fe_3_O_4_ NPs showed antimicrobial activity against *E. coli* (DIZ: 18.0 mm, minimum inhibitory concentration (MIC): 0.625 µg/mL), *B. subtilis* (DIZ: 17.0 mm, MIC: 0.625 µg/mL), and *C. albicans* (DIZ: 14.2 mm, MIC: 1.25 µg/mL). Similarly, Saedi et al. [138] showed that the carrageenan films incorporating Fe_3_O_4_-NH_2_-Ag had the strongest antimicrobial activity against *L. monocytogenes*, destroying the entire bacterial population in less than 3 h. However, Carr/Fe_3_O_4_-Ag prevented bacterial growth after 12 h. This was because Fe_3_O_4_ NPs were the best candidate for carrying Ag+ [139], and modification with amine-containing materials enhances their carrying capacity by binding Ag+ ions through the lone-pair bonding of their nitrogen atoms, which act as chelating agents [140]. Therefore, the bacterial population adsorbed more Ag+ compared to other films, and the amine functional groups also accelerated the attachment of NPs to the bacterial surface [141], thus enhancing the destruction of the bacterial cell wall.

Zeolites are tetrahedral atoms composed of crystalline metal oxides (e.g., Si, P, Al, Ti, B, Ga, Ge, Fe, etc.) [142], which have been considered to be safe and non-toxic for humans by the FDA [143]. Zeolites are crystalline hydrated aluminosilicates with a unique 3D structural composition consisting of interconnected cage-like structures or cavities, which allows them to complex molecular adsorption (e.g., histamine, water, and ethylene) and cation exchange [143]. Therefore, zeolites have been used as an active-scavenging packaging material to extend the shelf-life of fresh produce and meat products [143]. Hanula et al. [143] showed that the active packages prepared by adding zeolite (clinoptilolite) and aҫai extracts could prolong the shelf-life of *A. bisporus* mushrooms by reducing their water loss and slowing down the browning process both externally and internally in the mushrooms stored at 4 °C for 28 days after harvest. Consistently, Marzano-Barreda et al. [144] also reported that active packaging of zeolite NPs in a polymer matrix of PBAT, citric acid, and cassava starch reduced the metabolism of fresh broccoli florets over 7 days while keeping color and vitamin C content unchanged.

### 3.4. EO-Loaded Nanoemulsion 

Essential oils are secondary metabolites of plants and contain antimicrobial components such as polyphenols, terpene compounds (monoterpenes, sesquiterpenes, and diterpenes), and aldehydes [145]. 

Nanocapsules comprise a liquid or solid core surrounded by a polymeric coating which completely isolates the encapsulated contents from the environment and serves two functions, i.e., (1) providing a barrier to prevent factors such as oxidation and hydrolysis from destroying the encapsulated material in the nanocapsule and (2) controlling the release of active compounds [57]. For example, Khachatryan et al. [146] developed nanocapsules encapsulating propolis extract in a biodegradable natural polymer matrix, thereby controlling the release of antimicrobial propolis components. Nanoemulsions are also colloidal, kinetically stable systems characterized by very small droplet sizes ranging from 10 nm to 1000 nm in oil-in-water emulsions containing solid spheres with amorphous and lipophilic surfaces [147]. EO nanoemulsions provide a uniform distribution of partially or fully hydrophobic components in the hydrophilic matrix. This encapsulation system facilitates the controlled release of antimicrobial agents and enhances the antimicrobial properties of the nanocomposite film [148].

The hydrophobicity of EO nanoemulsions can alter bacterial membrane structure, especially unsaturated fatty acids [148]. Phenolic compounds in EO can act as protonophores, transporting protons across the lipid bilayer and resulting in proton kinetic dissipation [109]. Terpenoids cause loss of cell membrane integrity and proton dynamic dissipation [149]. High concentrations of aldehydes prevent bacterial cell division, while low concentrations of aldehydes block cell division interactions [150]. Importantly, the antimicrobial activity of EOs can be enhanced by encapsulation with a variety of NPs (e.g., liposomes, polymeric NPs, and nanoemulsions), where the nanomaterials form an external nanocapsule while the internal core is EOs. These two active agents can engage in surface disruption and energy balance disruption through complementary actions, thus combating different pathogens by different mechanisms [147]. 

### 3.5. Brief Summary of Antimicrobial Activity of Active Food Packaging 

Lipid oxidation is related to methemoglobin formation and meat discoloration [151], while weight loss is related to the loss of dry matter and moisture in fresh produce due to transpiration and postharvest respiration [152]. Both lipid oxidation and weight loss gradually increase with longer storage time. Therefore, reducing lipid oxidation and weight loss can contribute to extending shelf-life. 

Analyzing other relevant studies in Table 2, NPs could extend the shelf-life of food products by lowering lipid oxidation by an average of ~350.74% and weight loss by ~28.39% during the longest storage period compared to pure film (Figure 4A). However, the effectiveness of NPs in extending shelf-life may be influenced by the type of NPs used, the polymer–NPs interactions, the food product, and storage conditions. More research analyses are needed to fully understand the potential benefits of NPs in food preservation.

Additionally, the incorporation of NPs into the polymer films showed a significantly greater antibacterial efficacy against *S. aureus* compared to the neat polymer films (*p* = 0.034) (Figure 4B). The differences in the antibacterial activity of the nanocomposite films against Gram-negative *A. niger* and *E. coli*, and Gram-positive *L. monocytogenes* and *S. aureus* can be explained by the variation in the initial bacterial loading, the dispersion of the tested bacteria on the food surface, the bacterial susceptibility to the NPs, and the concentration of NPs in the nanocomposite films [153].

**Table 2 foods-13-02014-t002:** The antimicrobial properties of metal-based NPs in active packaging.

Nanofillers		Polymeric Matrix	Packaging Form	Findings with Optimal NPs Compared to Non-NPs	Application Product	References
Concentration/Percentage (%) Weight of Optimal Metal-Based NPs	Types of Metal-Based NPs					
2%	Ag NPs	Agar	Film	Prevent direct oxygen contact by up to 15 days. Maintain color. Retard oxidative rancidity.	Fresh beef loin	[115]
2%	Ag NPs	Pullulan-curcumin	Edible film	Show a greater antioxidant activity. Maintain the textural and physicochemical meat attributes up to 14 days.	Broiler breast	[154]
0.25%	Ag NPs	HPMC	Film	Inhibit the proliferation of *C. gloeosporioides* at room temperature (20 °C) up to 14 days. Maintain the physiological functions and quality of papaya during storage.	Papaya (*Carica papaya* L.)	[155]
10%	Ag NPs	Cellulose	Packets	Prevent the growth of A. hydrophila at room temperature (25 ± 2 °C) up to 7 days. Enhance shelf-life with no significant changes in nutritional values and moisture content.	Cabbages and tomatoes	[156]
5%	Ag NPs	PLA	Film	Effectively reduce vitamin C loss. Delay the decline of total phenols and 1-Diphenyl-2-picrylhydrazyl (DPPH) for 2 days. Retard the consumption of acid in strawberry physiological metabolic activities.	Strawberry	[157]
5%	Ag NPs (with 10% EOs)	Chitosan	Film	Increase shelf-life up to 12 days.	Strawberry	[158]
10%	ZnO NPs	PBAT	Film	Increase antimicrobial effects against both *E. coli* (DIZ: 14.1 mm) and *S. aureus* (DIZ: 15.1 mm)	N/A	[159]
12.5%	ZnO NPs	Gelatin/Tapioca starch	Film	Increase antibacterial effects against both *S. aureus* ZOI: 67.28 mm^2^) and *E. coli* (ZOI: 85.30 mm^2^).	N/A	[123]
1.5%	ZnO NPs	Pectin	Film	Increase antibacterial rates against both *E. coli* by 97.2%, and *S. aureus* by 98.3%.	N/A	[160]
4 mL	ZnO NPs	HPS/PVA/PA	Film mats	Increase antimicrobial activity against *E. coli*, *S. aureus*, *F. oxysporum*, *A. niger, P. expansum*, and *A. flavus*, with DIZ of 15.1, 12.1, 12.3, 16.0, 18.0, and 22.0 mm, respectively.	N/A	[124]
0.5 g	ZnO NPs	CMC	Film	Enhance antifungal activity (*A. niger*) by ~1.4 cm colony diameter. Inhibit the physiological and metabolic activity of fruit during postharvest storage.	Cherry tomatoes	[161]
100 mg	Cinnamaldehyde					
1.5%	ZnO NPs	PLA	Film	Increase the shelf-life up to 16 days. PLA/ZnO NPs/ZEO increased antibacterial activity against *S. aureus* (ZOI: 691 mm^2^), *E. coli* (ZOI: 200.67 mm^2^), *B. cereus* (ZOI: 513.33 mm^2^) and *P. aeruginosa* (ZOI: 78.33 mm^2^), and antioxidant activity by 69.14%. PLA/ZnO NPs/MEO increased antibacterial activity against *S. aureus* (ZOI: 513.33 mm^2^), *E. coli* (ZOI: 113.28 mm^2^), *B. cereus* (ZOI: 314.33 mm^2^) and *P. aeruginosa* (ZOI: 63.56 mm^2^), and antioxidant activity by 49.08%.	Fresh fish fillets	[162]
1.5%	ZEO					
1.5%	MEO					
0.8 g	Ag NPs	Starch/PBAT	Film	Have synergistic effects on inhibiting the growth of *S. aureus* and *E. coli.*	Peaches and nectarines	[163]
0.2 g	ZnO NPs					
0.99 g in 15 mL water	ZnO NPs	Chitosan	Film	Inhibit growth of *E. coli*, *S. aureus*, and *C. albicans*, with DIZ over ap-prox. 30 mm, approx. 30 mm and over approx. 20 mm, respectively. Keep an acceptable visual appearance at room temperature (30 °C and 60% relative humidity) for 14 days.	Grape	[164]
15 mL	Ag NPs					
1 mL	citronella EO					
0.015%	TiO_2_ NPs	PLA	Film	Elevate antimicrobial activity against *E. coli* (DIZ: 15 mm) and *S. aureus* (DIZ: 18 mm). Increase antioxidant activity.	N/A	[165]
3%	Lycopene					
5%	TiO_2_ NPs	Alginate and *aloe vera*	Edible coating	Inhibit the growth of *S. aureus* (DIZ: 21.89 mm), *E. coli* (DIZ: 17.98 mm), and A. fumigatus (DIZ: 21.29 mm). Prolong the shelf-life with minimal mass loss during storage up to 16 days.	Tomato	[166]
0.003 g	TiO_2_ NPs	Starch/PVA	Film	Extend shelf-life without sign of microbial infection (*S. aureus*) up to 22 days.	Cherry tomato	[167]
0.4 g	elderberry extract					
1%	TiO_2_ NPs	Chitosan	Film	Reduce total bacterial count, *E. coli*, and *S. aureus* after 10 days of storage. Maintain color, taste, and odor. Reduce forming volatile compounds, including ammonia, trimethylamine, dimethylamine and methylamine.	Minced meat	[129]
1.5%	*Cymbopogon citratus* EO					
0.3%	Ag NPs	Chitosan/polyethylene oxide	Films	Inhibit the activity of *E. coli*, *S. aureus*, *C. Albicans*, and *A. niger*, with DIZ of 8, 11, 21, and 23 mm, respectively.	N/A	[168]
0.8%	TiO_2_ NPs					
1%	Ag NPs	PLA	Films	Delay the loss of firmness, titratable acid, and vitamin C during storage. Extend the post-harvest life of mangoes to 15 days.	Fresh mango	[169]
2%	TiO_2_ NPs					
9%	Bergamot EO					
3%	CuO NPs	Starch	Film	Reduce colony counts of *S. aureus* and *E. coli*.	N/A	[133]
N/A	Au NPs	PVA/glyoxal	Film	Enhance antimicrobial activity against *E. coli* (DIZ: 13 mm). Extend shelf-life up to 5 days.	Banana	[135]
10%	Fe_3_O_4_ NPs	Chitosan/pectin	Film	Inhibit the growth of *S. epidermidis* and *E. coli*.	N/A	[170]

Note: HPMC: hydroxypropyl methylcellulose; PLA: polylactic acid; EO: essential oil; PBAT: poly(butylene adipate-co-terephthalate); PVA: polyvinyl alcohol; HPS: hydroxypropyl starch; PA: palmitic acid; N/A: used as food packaging, but not specifically for foodstuffs; CMC: carboxymethylcellulose; ZEO: Zataria multiflora EO; MEO: Menthe piperita EO; DIZ: diameter of inhibition zone; ZOI: zone of inhibition. This table includes the parameters for the optimal metal-based NPs.

## 4. Use of Nanofillers in Intelligent Food Packaging

### 4.1. The Definition of Intelligent Packaging 

Intelligent packaging can communicate with consumers, monitor, track, record, and convey external or internal changes that occur in the product or its environment without affecting the food along the food supply chain [25]. The intelligent packaging is mainly categorized into indicators, sensors and data carriers [171]. For example, Figure 5a shows the time–temperature indicator (TTI), providing consumers with information on the food quality at specific temperatures and times. Figure 5b presents a freshness indicator that detect changes within food packages, including microbial growth and metabolite levels, and further indicate spoilage or degradation of the food. Figure 5c represents a pH sensor designed for the quick measurement of pH levels and food quality. Figure 5d is radio-frequency identification tags (RFID), providing information on storage, transport, distribution, and sale. 

### 4.2. Nanocomposites-Based Intelligent Films 

#### 4.2.1. Anthocyanins and Their Utility

Colorimetric nanocomposite-based intelligent packaging are increasingly being applied to monitor the freshness and assess the quality and safety of perishable foods, fresh produce, and meat/seafood. It is designed to induce color/ammonia changes in response to changes in environmental pH during food spoilage, which can be achieved by adding natural dye. Since they offers naturalness, non-toxicity, and biodegradability compared to synthetic dyes that may have toxicity and contamination concerns [172]. Among the various natural dyes, anthocyanins (AN) are particularly pH-responsive and can therefore be used to detect the presence of leakage (e.g., oxygen exposure) and to indicate microbiological safety by detecting the volatile nitrogen compounds and amines from spoiled products and microbiological contamination [173]. However, incorporating AN into food packaging generally suffers from some shortcomings, including a time-consuming and costly extraction process and insufficient stability [174,175]. Zheng et al. [172] reported that CNCs have negatively charged sulphonic acid groups on their surface, derived from sulfuric acid hydrolysis. They could form ionic bonds with flavonoid cations in AN, thus avoiding nucleophilic attack through water to modulate AN’s indicative properties. This ionic bond could also change dynamically with the pH change. Thus, this chitosan/AN intelligent packaging enhanced by CNCs intuitively monitors shrimp freshness and slows down the spoilage process. The multifunctional bio-nanocomposite films (KCZ-MAE) prepared from konjac glucomannan (KGM)/chitosan (KC) with Nano-ZnO NPs and mulberry anthocyanin extract (MAE) also exhibited relatively large color changes from red (pH = 2–4) to blue (pH = 10–12) in different buffer solutions, serving as a freshness indicator for chicken millets [176]. Consistently, Wang et al. [177] also incorporated pH-responsive AN from eggplant peel extract into the chitosan/esterified chitin nanofibers (CN) polymer matrix. It was sensitive to pH 3–11 buffer solution and possessed suitable ammonia and acid sensitivity, making it suitable for detecting pork spoilage. Furthermore, it is also a trend to use hybrid pigments in packaging materials. Duan et al. [178] incorporated curcumin and AN dye pigment into electrospun Lapland/chitin nanofibers (379.07 ± 100.14 nm) as indicators of freshness of *Plectorhynchus cinctus* fish, i.e., pink for fresh products and pinkish blue for deteriorated products. 

AN phenolic compound can be applied to TTIs due to their temperature-sensitive nature [179]. The structures of AN change upon thermal degradation, which is influenced by the severity and conditions of heating [179]. Rachmelia and Imawan [180] developed a TTI label by incorporating ANs extracted from black maize into a chitosan polymer matrix, which showed the fastest color change from violet to blue to yellow at the higher temperature of 40 °C, whereas the slowest color change was at the lower temperature of 10 °C. Consistently, Amiri et al. [181] fabricated a TTI based on paraffin wax film containing black carrot ANs to monitor fish products. There was no change in color after 48 h of exposure at temperatures of −5 °C and 25 °C, whereas significant color changes were observed at higher temperatures (15 and 25 °C), which also corresponded to a significant increase in Thiobarbituric acid reactive substances (TBARS) and total volatile basic nitrogen (TVB-N), as well as a decrease in organoleptic parameters. Eskandarabadi et al. [182] also stabilized ANs extracted from red cabbage on MMT as a natural TTI for the active intelligent packaging film. 

#### 4.2.2. Other Compounds and Their Application 

Curcumin is a diphenolic hydrophobic compound derived from the roots of turmeric [183]. Wu et al. [184] reported that the release rate of curcumin at low pH (~2) was higher than that at neutral pH 6 and 7.4, respectively, suggesting that curcumin is a pH-sensitive compound to inform consumers about the food freshness. Salarbashi et al. [185] reported that soluble soybean polysaccharides/15 wt% SiO_2_ NPs/curcumin nanocomposite films changed color from yellow, orange to orange-red upon exposure to acidic, neutral, and alkaline media, respectively, revealing their application in detecting pH changes during shrimp spoilage.

Carbon quantum dots (CDs), known as fluorescent carbons, can be used as a fluorescent “on/off” sensor for detecting relative humidity (RH) [186]. Rahman and Chowdhury [187] showed that increasing the bread’s RH resulted in the quenching of the fluorescence of the nanocomposite film containing CDs due to the hydrogen bonding of water molecules from the wetted bread with the nanocomposite film.

Additionally, Zhai et al. [188] developed a novel colourimetric hydrogen sulfide (H_2_S) sensor based on gellan gum-capp Ag NPs for real-time monitoring of meat spoilage, since Ag has a superb ability to bind to H_2_S, forming Ag_2_S [188]. This gas sensor could analyse H_2_S with a detection limit of 0.18 μM at pH 7 and showed excellent selectivity for H_2_S against other volatile components produced from chicken breast and silver carp during spoilage, showing a visible color change from yellow to colorless [188]. Consistently, Kwon and Ko [189] also developed CNC-AgNP composites as efficient colourimetric freshness indicators for poultry products or broccoli.

## 5. Use of Nanofillers in Photodegradable and Biodegradable Food Packaging

### 5.1. Photodegradable Food Packaging

Photodegradation is the ROS formation on the surface of the photocatalytic materials (e.g., TiO_2_) by UV radiation, which promotes polymer chain breakdown and accelerates chain breakage [190]. Therefore, Goñi-Ciaurriz et al. [191] showed that under ~24 h UV exposure with 1.0 mW/cm^2^ light intensity, the incorporation of 5 wt% TiO_2_ NPs decreased the hydrogen-bonding elements of ethylene–PVA copolymers (EVOH) matrix, thereby promoting the photo-oxidative degradation (i.e., discoloration) of nanocomposite films, with total color variations (∆E) greater than 20, whereas neat EVOH polymer only had ∆E less than 1. Furthermore, Goudarzi and Shahabi-Ghahfarrokhi [192] developed starch/TiO_2_ (3 wt%) bio-nanocomposites using photochemical reactions, showing photo-degradability around 20% better than virgin starch/TiO_2_ films. Masoumeh et al. [190] further coupled TiO_2_ with transition metals, i.e., Fe_3_O_4_, which lowered the band gap of TiO_2_ and improved its photocatalytic activity at higher wavelengths of the electromagnetic spectrum. They have reported that the addition of 10 wt% Fe_3_O_4_/TiO_2_ increased the photodegradation of the starch nanocomposite film by approx. 232.17% as compared to neat polymers at 0–2 days of UV-A exposure timespan. 

### 5.2. Biodegradable Food Packaging

Biodegradation of biopolymers undergoes two processes, i.e., conversion from polymer to monomer and the mineralization of monomers into CO_2_, H_2_O, and biomass by microbial bio-assimilation. Large biopolymers are digested by extracellular enzymes in microorganisms (e.g., bacteria, fungi), while small molecules are transported to microorganisms to undergo endoenzymatic digestion [193]. Therefore, there are two pathways to alter the biodegradation mechanism, namely altering the crystallinity of the polymetallic matrix and changing the microbial biodegradation pathway [194]. 

The addition of CNCs reduced the biodegradation rate of the PVA–gelatin matrix after 28 days of soil burial test, with the lowest weight loss (i.e., 12.58%) for the 5 wt% addition, compared to 12.62% and 12.9% for 10 wt% and unreinforced films, respectively. This may be because the 5 wt% CNCs can be better dispersed in the PVA–gelatin matrix to create a better 3D network structure, increasing the physical integrity of the composites and reducing the biodegradation rate [195]. Similarly, the increased binding of chitosan–cellulose phthalate acetate films because of an increase in the concentration of ZnO NPs from 2 wt% to 7.5 wt%, resulted in an extended degradation time of about 5 months, whereas the pure chitosan films were biodegraded within 4 weeks [196]. Yeasmin, Yeum, and Yang [197] also assessed the biodegradation of pullulan (PULL), tempo cellulose nanofibrils (TOCNs), and MMT nanocomposite films after 4 and 18 days using soil burial tests. They found that the addition of MMT decreased the biodegradation rate because strong hydrogen bonds were formed between the hydroxyl groups of the PULL matrix and the MMT, thereby increasing the cohesion of the PULL matrix and reducing its water sensitivity. 

Furthermore, Oliver-Ortega et al. [198] reported that while the biodegradation process seems to follow the same pathway, i.e., the crystallinity of the nanocomposite increases from aging, which whitens the nanocomposite and decreases the edge mass. The incorporation of 2 wt% and 4 wt% of nanoclay (Bentonite) extended the biodegradation time of PLA nanocomposites to 11 and 15 days, respectively, while the biodegradation of the pure polymer time was about 6 days. Enzymatic biodegradation of nanocomposite films begins at the surface, suggesting that enzymes may attach to the surface of the nanoclay. Therefore, the biodegradation process may be more susceptible to inhibition by the nanoclay [198]. Similarly, the incorporation of 0.05% AgNPs reduced the biodegradation rate of chitosan–gelatin nanocomposite films by about 9% after 14 days of exposure to the soil as compared to the pure polymer [199]. This may be attributed to the antimicrobial activity of Ag NPs against soil microorganisms, inhibiting their attack on the polymer chains weakened by the infiltration of soil moisture into the polymer network [199]. Furthermore, Perera et al. [200] showed that the biodegradation of chitosan–alginate nanocomposite film was significantly decreased by 10.95% during 3 months with increasing TiO_2_ NPs concentration (0.1 wt%: 100%; 0.3 wt%: 89.06% ±  1.04) due to the antimicrobial properties of TiO_2_ NPs.

However, Luo, Lin, and Guo [201] showed that TiO_2_-containing films started the biodegradation stage earlier and produced a higher percentage of CO_2_ after 80 days of incubation compared to pure PLA films, as indicated by biodegradation percentages being 78.9% for 0% TiO_2_, 86.9% for 1% TiO_2_, 92.0% for 2% TiO_2,_ and 97.8% for 5% TiO_2_, respectively. The uniformly dispersed TiO_2_ in the PLA matrix resulted in an easier penetration of water molecules to the polymer matrix, then initiating the biodegradation process by microorganisms. Similarly, the biodegradation phase of films containing organo-modified MMT Halloysite nanotubes and Laponite^®^ RD also began earlier than that of pure PLA films [202]. Consistently, increasing carbon NPs concentration also increased the weight loss behavior of PBAT composites, e.g., the composite films containing 3wt% CNPs and 5wt% CNPs degraded faster compared to pure PBAT, achieving 43.4% and 48.6% of weight loss after 8 weeks, respectively [10]. 

The above comparative results suggest that the concentration and type of nanofillers incorporated into biodegradable polymers should be balanced with the purpose of the food packaging system. If the main purpose is to improve mechanical, barrier, and antimicrobial properties by considering factors such as polymer–nanofiller compatibility, permeability, and microbial interactions, the lower biodegradation rate may be compensated. Therefore, nanocomposite systems need to be carefully designed and optimized to achieve the desired biodegradation properties.

## 6. Migration Process 

Migration is a mass transfer process whereby low molecular mass components initially presenting in the food packaging (high concentration) are released into the contained product (e.g., food or beverage) (low concentration) and eventually reach equilibrium. This process is subjected to diffusion (diffusion coefficient (D), Equation (2)) and adsorption (partition coefficient ($k_{P})$, Equation (4)), respectively [203].
(4)$$\frac{x}{m}=k_{p}\times \frac{1}{n}$$

x: mass of adsorbed air; m: adsorbed volume at pressure of P; k_p_ and n: constants.

The development of nanocomposites may result in unintentional migration of nanofillers into food products, raising safety concerns for consumers. Therefore, when developing new food contact materials (FCM), overall and specific migration tests must be performed using food simulants that can mimic food behaviors. There are six common food simulants, including ethanol 10% *v*/*v*, acetic acid 3% *v*/*v*, ethanol 20% *v*/*v*, ethanol 50% *v*/*v*, vegetable oil, and poly-2,6-diphenyl-p-phenylene oxide [6,203]. 

### 6.1. Factors Affecting the Migration Process

The initial concentration of migrants in the polymer and types of foods/food simulants should be considered for migration calculations. For example, Bott, Störmer, and Franz [21] reported that after 10 days at 60 °C, the total silver migration concentration of Ag NPs at concentrations of 50, 150, and 250 mg kg^−1^ was 2.4 µg dm^2^, 13.2 µg dm^2^ and 115.1 µg dm^2^ in 10% ethanol and 168.5 µg dm^2^, 444.8 µg dm^2^, and 1010.9 µg dm^2^ in 3% acetic acid, respectively. However, no silver was released from the low-density polyethylene (LDPE) polymer matrix containing any concentrations of Ag NPs, even after 10 days in 95% ethanol at 60 °C and 24 h in isooctane at 40 °C.

The D demonstrates the kinetics of migration within a polymer matrix or foodstuffs, quite being affected by temperature and time. Extreme temperature fluctuations (e.g., from freezing to cooking temperatures) can increase the D and migration activity of the chemical in the package by 6- to 7-fold, which in turn allows the chemical to overcome the attraction of the surrounding molecules and separate from the polymer matrix [204]. Therefore, each packaging material is only suitable for one contact temperature. Furthermore, the shorter the distance between the material and the foodstuff, the faster and easier the diffusion of migrants will be, as less time will be spent [6]. Moreover, according to Equation (2), D is also mainly a function of the size or the free cross-section of the migrated NPs (measured in molecular weight, i.e., M_r,i_), which decreases exponentially with increasing migrant size [205]. Bott, Störmer, and Franz [22] modelled the migration of spherical carbon NPs from LDPE (10 days at 40 °C) as a function of size and sphere diameter and showed that carbon NPs with a maximum diameter of 4 or 5 nm could potentially migrate (D_LDPE_: 1.6 × 10^−20^ and 2.1 × 10^−22^ cm^2^ s^−1^, respectively). If the concentration of 5 nm particles would migrate, it should require 25,000 mg kg^−1^ and consists of all 5 nm particles, which is completely unrealistic and therefore may never occur in FCM plastics. In summary, the modelled migration rate decreases exponentially with increasing diameter and the chances of a consumer being exposed to nanoparticles from FCM plastics are negligible and, in any case, cannot be measured analytically.

The $K_{P}$ describes the relative solubility and concentration of the migrant in the polymer matrix and food at equilibrium. The decrease in $K_{P}$ is related to the reduction in molecular weight of nanofillers and the dynamic viscosity of polymers, which may increase the migration process [206]. Moreover, the polarity of migrated molecules and food simulants impacts the $K_{P}$. The amount of migration from the multilayer polyamide to 3% acetic acid was higher than for other food simulants because they have similar polarity [207]. Toluene showed the best migration behaviour from LDPE and PP in isooctane compared to other stimulants such as 50% ethanol, 3% acetic acid, and 10% ethanol, as toluene is only weakly polar and isooctane is the only non-polar simulant [208]. 

### 6.2. Assessment Methods for Nanofillers

To characterize the nanocomposite structure, it is necessary to consider the dispersion of the nanofillers, the variation in the mass matrix, and the type of nanofiller–polymer interface. Given the complexity of nanocomposites and their small proportion of applications in food packaging systems, detecting the migration of nanofillers into food products requires more sensitive analytical techniques [209]. The independent methods cannot provide all the required information about the concentration, composition, and physicochemical properties of nanofillers in complex matrices [209]. There are only a few synthetic methods available for the efficient detection of nanofiller.

Scanning electron microscopy (SEM), transmission electron microscopy (TEM), and exclusion chromatography (SEC) could segregate nanofillers down to 1 nm, facilitating directly studying the size, shape, structure, density, dispersion, and coagulation of nanofillers in complex solid samples [6,24]. However, these methods are quite time-consuming as they require the counting of more than hundreds of nanofillers to obtain comprehensive and adequate information. Meanwhile, these methods are destructive, indicating that the verification tests could not be conducted on the same sample [24].

Inductively Coupled Plasma and Mass Spectrometry (ICP-MS) can evaluate dissolved samples and is highly selective, sensitive, and accurate. It can rapidly quantify the elemental composition of spherically structured NPs down to 0.1 to 10 ppm [203]. A digestion process is required before injecting the nanofillers into the ICP [210]. When combined with Laser ablation, i.e., LA ICP-MS, the determination limits for Ag and Cu could be adjusted from 0.03 mg/kg to <0.08 mg/kg and from 0.01 mg/kg to <4 mg/kg, respectively [211]. Moreover, the NP Tracking Analysis (NTA) and Dynamic Light Scattering (DLS) are recommended for suspension samples [203].

Furthermore, both wide-angle X-ray diffraction (XRD) (WAXS) and small-angle XRD (SAXS) can assess solid samples. They are not destructive, and do not require sample pretreatments. Thus, they can be used to assess the elemental composition or crystalline arrangement of nanomaterials, as well as the form of different nanocomposites, such as aggregation, intercalation, exfoliation, and dispersion. The other spectroscopic technique, e.g., ultraviolet–visible spectroscopy (UV–VIS), is generally used to obtain information such as the presence and characterization of nanomaterials due to its low costs and easy usage [212].

These analytical methods are essential to ensure the safety and regulatory compliance of nanocomposites used in food packaging. Additionally, the (re)assessment of nanotoxicology, exposure estimates, and harmful effects on humans are required when additional studies on the toxicological assessment of nanocomposites are feasible [24]. 

### 6.3. Application of Nanofiller to Reduce Migration

However, the incorporation of several nanofillers in polymers may reduce the migration process of chemicals from FCM to food as follows.

Migrants tend to choose paths with the least diffusion resistance in the polymer chain, usually avoiding the positions occupied by NPs, resulting in longer paths [213]. The volume fraction of NPs, their orientation concerning the diffusion direction, their shape and L/D ratio, and their dispersion degree could increase the tortuosity of the diffusion path and reduce the overall diffusion rate in the nanocomposites [214]. For instance, Dardmeh et al. [214] showed that the addition of 3 wt% Cloisite 15A to the PET matrix effectively reduced the migration of terephthalic acid (TPA) under storage conditions because the clay was uniformly dispersed in the PET matrix and formed exfoliated structures. Consistently, de Abreu et al. [215] reported that Cloisite 30B nanoclays slowed down the migration of caprolactam, 5-Chloro-2-(2,4-dichlorophenoxy)phenol (triclosan), and trans,trans-1,4-diphenyl-1,3-butadiene (DPBD) from polyamide nanocomposites to food simulants up to six times. Garofalo et al. [216] also showed that PA/PE multilayer nanocomposite films could overcome the possible migration of NPs by using a functional barrier between the nanomaterial and the food. Similarly, Seray and Hadj-Hamou [217] also reported that the simultaneous use of these two nanofillers (i.e., ZnO NPs, Cloisite nanoclay) in the PBAT matrix could reduce Zn^2+^ release due to the need for Zn^2+^ to diffuse gradually in the increased tortuosity pathways formed by the dispersed Cloisite. This provides a suitable candidate for the application of sustained and controlled release of Zn^2+^ in active food packaging, which improves the inhibition of bacterial growth and prolong the freshness and quality of packaged food products compared to the use of ZnO NPs alone.

Furthermore, NPs act as heterogeneous nucleating agents in the polymer matrix, thereby increasing crystallinity and reducing chemical migration. Farhoodi et al. [218] reported that based on differential scanning calorimetry (DSC) analyses, the incorporation of TiO_2_ NPs (3 wt%) into the polyethylene terephthalate (PET) matrix resulted in a substantial increase in the final crystallinity of the PET polymers ($X_{C}$ = 39.68%) compared to neat PET ($X_{C}$ = 33.48%). The results of gas chromatography showed that the migration of ethylene glycol from the nanocomposites was lower than that from neat PET [218].

## 7. Toxicological Effects of Nanofillers

The toxicity concerns of the migrated nanofillers involve cytotoxicity and ecotoxicity.

### 7.1. The Cytotoxicity of Migrated Nanofillers

Free NPs can penetrate the biological environment and inevitably come into contact with various biomolecules (proteins, sugars, and lipids) dissolved in body fluids such as interstitial fluids, lymph, or plasma, leading to potential cytotoxic effects [219]. Meanwhile, nanofillers smaller than 6 nm are easily excreted through the kidneys, while those larger than 200 nm accumulate in the liver and spleen [220] (Figure 6). 

The smaller nanofillers are more toxic than larger ones since they are more readily and quickly absorbed and distributed by the organs (e.g., stomach and lungs). The wider surface area compared to the total mass of the smaller nanofillers can also generate more ROS, which can cause damage and malfunction to organelles, such as mitochondria and nuclei, leading to apoptosis and cell death [219]. 

The hydrophilic and cationic nanofillers are more toxic than hydrophobic, neutral, or anionic ones, partly due to their strong affinity for negatively charged plasma membranes. Therefore, their circulation and residence time in the blood vessels is prolonged, increasing the risk of blood clotting and cardiovascular disease [56]. 

The rods could be mostly uptake and have a longer blood half-life as compared to that in spheres, cylinders, and cubes [221]. Nevertheless, Rozhina et al. [222] found that carbon nanomaterials were more toxic and genotoxic for cells than nanoclays, regardless of shape. 

Dermal contact, inhalation, and oral ingestion are the three main routes of human exposure to NPs, with oral exposure being the most common [223] (Figure 6). 

Children are more vulnerable to toxins. They absorb larger doses of pollutants per unit of body weight compared to adults, since the detoxification organ systems are not fully grown and developed [56]. Similarly, Wang et al. [224] orally administrated TiO_2_ NPs (~75 nm) to young rats (3 weeks) and adult rats (8 weeks) at doses of 0, 10, 50, and 200 mg kg^−1^ bw/d for 30 days. The liver edema was only evident in the young rats treated with 50 and 200 mg kg^−1^ bw TiO_2_ NPs, but not in adult rats. Adult rats only showed inflammatory cell infiltration in the 10 and 50 mg kg^− 1^ bw TiO_2_ NPs-treated groups.

Additionally, the cytotoxic risk is likely to increase if the daily intake of NPs migrated from FCM (Daily Dietary Index: DDI, Equation (5)) exceeds the Reference Dose (RfD) (e.g., a maximum migration of 0.05 mg/kg for some NPs by legislation [24]), resulting in a Health Risk Index (HRI, Equation (6) [225]) greater than 1. Meanwhile, NPs typically need to enter the cell membrane through penetration or endocytosis, taking time to exert cytotoxic effects. Therefore, the cytotoxic effects of migrated NPs can be influenced by the dosage and duration of exposure.
(5)$$DDI=A\times B\times \frac{C}{BW}$$

A: Concentration of NPs in food/migrate in food packaging (mg kg dry weight^−1^); B: daily intake of food (kg wet weight day^−1^); C: conversion factor (e.g., 0.085: weight of fresh vegetables converted to dry weight); BW: average human body mass (kg).
(6)$$HRI=\frac{DDI}{RfD}$$

However, the probability of NPs migration into food is small when NPs are fully incorporated or encapsulated in the polymer matrix. Meanwhile, appropriate surface treatments can modify the structure, size, solubility, behaviour of nanofillers and reduce their cytotoxic or mutagenic effects on cells or cellular components. Therefore, the possible health risks of migrated NPs from food packaging are still not clearly defined and require further research. 

In addition, legislation and regulation are also needed to protect consumers from adverse exposure risks, while developing effective regulatory provisions is challenged by limited scientific evidence and uncertainty.

### 7.2. The Ecotoxicity Effect of Nanofillers

Nanocomposites may cause contamination by releasing nanoscale compounds during degradation. Even low concentrations of ZnO NPs can compromise membrane integrity and reduce algal cell viability [226]. Ag NPs could inhibit embryonic development and destabilize lysosomal membranes of adult hepatopancreatic cells in the oyster *Crassostrea virginica* [226]. TiO_2_ NPs can reduce brood size and body length in large juvenile fish, disrupt their digestive enzymes (e.g., amylase and esterase), interfere with nutrient uptake and energy partitioning, and induce ROS production [226]. Therefore, the expanded use of NPs in the food packaging industry has raised significant environmental concerns. 

However, after the disposal of nanocomposites, several organisms in the environment might alter product properties, including photochemical transformation by light, biotransformation by microorganisms, and oxidation by oxygen. These may lead to a lesser environmental impact than starting materials [28]. 

Moreover, the final distribution of nanofillers after being discarded into the environment is not fully understood. Therefore, it is also difficult to determine whether nanocomposites ultimately contaminate and bioaccumulate the food chain and pose a risk to human health. More robust techniques are required to characterize the real behaviour of NPs.

## 8. Conclusions and Future Perspectives 

Compared to the previous reviews [142,193,227,228], this review discussed more comprehensively the improvement of mechanical, barrier, antimicrobial, photo-, and biodegradation properties of nanocomposites by inorganic and organic nanofillers in the last five years. It is also novel to summarize that the addition of EOs in nanoemulsions and nanohybrids can provide synergistic antimicrobial properties, help extend the shelf-life of packaged foods and ensure food quality. Although some studies [142,228,229] have reported that the migration process of nanofillers may cause cytotoxicity, this review used Bott’s equation (Equation (2)) to explain that the migration process is negligible, especially when fully incorporated or encapsulated in the host polymer matrix or surface modified. Furthermore, this review also innovatively discussed how the incorporation of some nanofillers into the polymer may reduce chemicals migration and achieve sustained controlled release of ions. However, the bioaccumulation of nanocomposites through the food chain is unknown as the final distribution of nanocomposites in the environment is unclear. 

Additionally, while some companies have successfully used nanoparticle-based polymeric materials in commercial food packaging, it remains a research challenge to scale up production from laboratory to commercial scale. It is time-consuming and costly to maintain consistent performance, ensure reproducibility, and meet regulatory requirements. While research continues, these factors affect the decision to add NPs in food packaging materials. Optimizing manufacturing technologies and costs, improving regulatory frameworks and building trust between the food industry, packaging manufacturers and consumers will help ensure safety and environmental sustainability of nanofillers.

## Figures and Tables

**Figure 1 foods-13-02014-f001:**
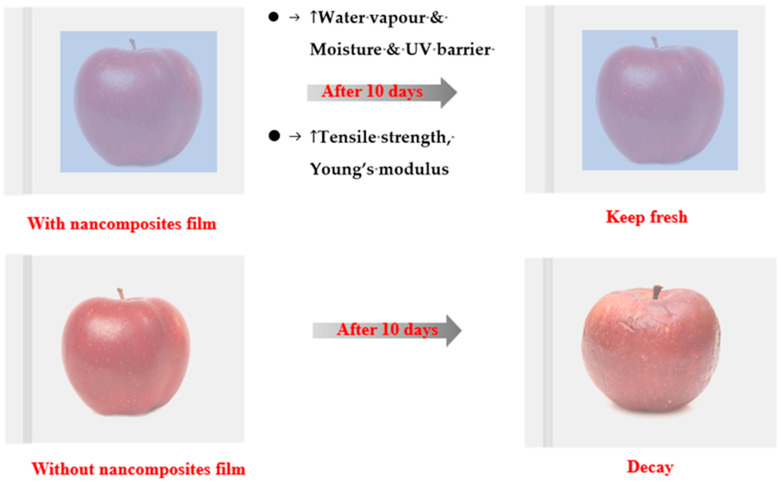
Schematic representation of the use of nanofillers to improve the physiochemical properties of food packaging, which keeps apples fresh after about 10 days of storage. Created with BioRender.com.

**Figure 2 foods-13-02014-f002:**
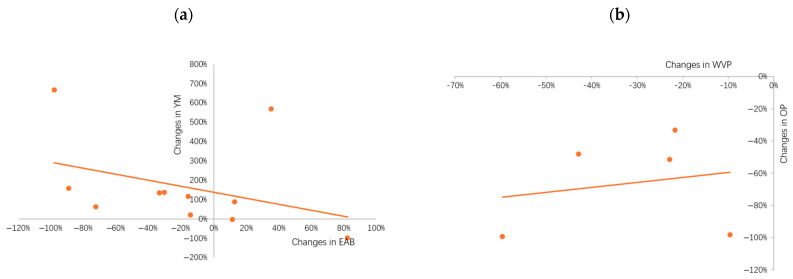
Changes in (**a**) elongation at break (EAB) vs. Young’s modulus (YM): y = −1.55x + 1.38, R^2^ = 0.128, r = −0.358, *p* = 0.018; and (**b**) changes in water vapor permeability (WVP) vs. oxygen permeability (OP): y = 0.30x − 0.57, R^2^ = 0.039, r = 0.197, *p* = 0.065, when adding nanofillers as compared to control. These correlations are obtained from the data in Table 1. r: Pearson correlation coefficient; R^2^: Coefficient of determination. The orange line is the regression line.

**Figure 3 foods-13-02014-f003:**
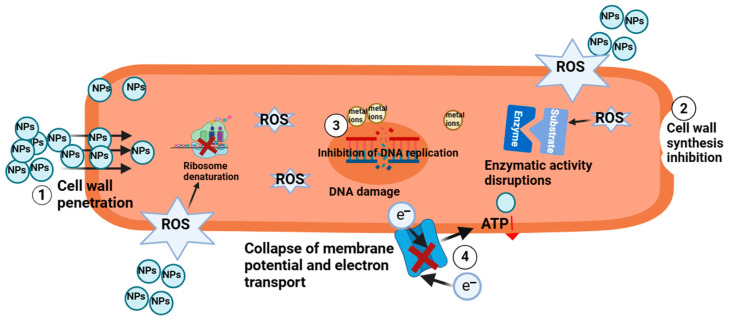
Mechanisms of antimicrobial activities of metal NPs. (1) Cell wall penetration; (2) the inhibition of cell wall synthesis; (3) the inhibition of DNA replication; (4) collapse of cell membrane potential and electron transport with a decrease in ATP. Created with BioRender.com.

**Figure 4 foods-13-02014-f004:**
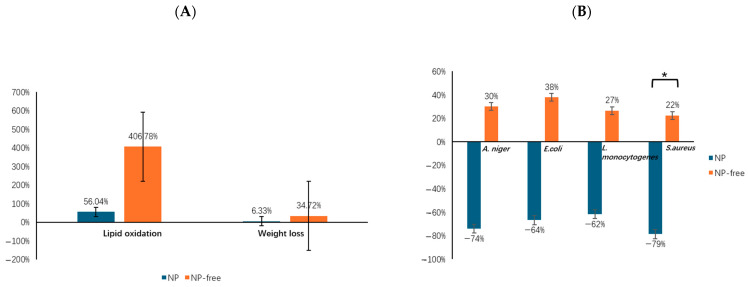
(**A**) Comparison of NP films and NP-free films for shelf-life extension applications, where shelf-life is indicated by lipid oxidation and weight loss. (**B**). Comparison of antimicrobial activity of NP films and NP-free films against Gram-negative *A. niger* and *E. coli*, and Gram-positive *L. monocytogenes* and *S. aureus*. The column indicates average changes in control film values and optimal film values for each parameter during the longest storage period, expressed as mean ± standard error. *: *p* < 0.05 between NP and NP-free film.

**Figure 5 foods-13-02014-f005:**
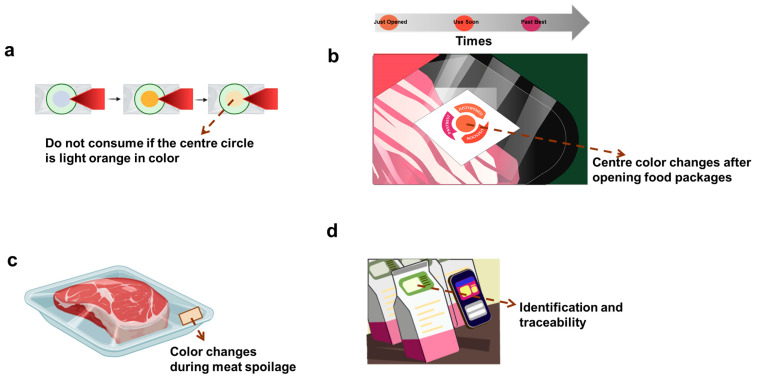
Commercial application of indicators in intelligent packaging systems. (**a**) Time and temperature indicators (TTI); (**b**). freshness indicators; (**c**) pH sensors; (**d**). RFID. Created with BioRender.com.

**Figure 6 foods-13-02014-f006:**
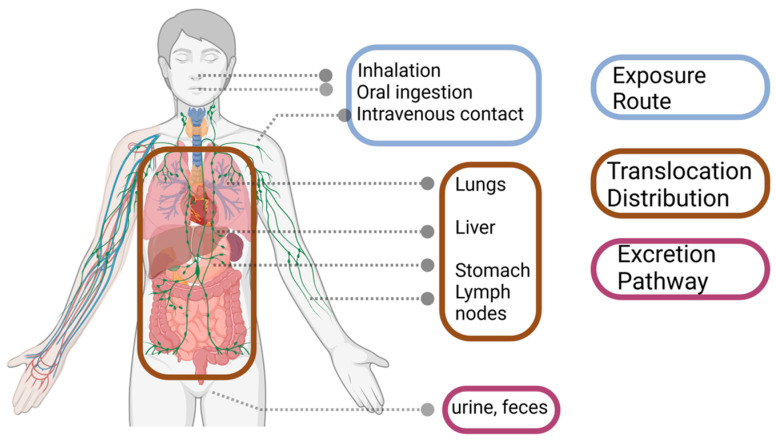
Exposure routes, translocation distribution, and excretion pathways of nanofillers in the human body. Created with BioRender.com.

**Table 1 foods-13-02014-t001:** Nanofillers are used in improved food packaging.

Study	Nanofillers	Polymer Matrix and Added Components	Mechanical Properties			Barrier Features			Thermal Characteristics
			TS (MPa)	EAB (%)	Young’s Modulus (MPa)	WVP (g m^−1^ Pa^−1^ s^−1^)	OP (cc Mil m^−2^ Day Atm)	Light Transmittance (%)	Mobility (Tg (°C)
[92]	MMT	PET, PAA	N/A	N/A	N/A	2.50 × 10^−12^ → 2.26 × 10^−12 ^ (−9.60%)	1.01 × 10^−16^ → 1.7 × 10^−18^ (−98.32%)	N/A	N/A
[93]	MMT	PVA boiled rice starch	16.8 → 33.5 (+99.40%)	257 → 4.6 (−98.21%)	290.0 → 2220.0 (+665.51%)	N/A	N/A	N/A	N/A
[94]	Halloysite nanotubes	PLA	22.5 → 37 (+64.44%)	2.1 → 1.4 (−33.33%)	1800.0 → 4200.0 (+ 133.33%)	1.62 × 10^−10^→ 1.27 × 10^−10^ (−21.60%)	1.06 × 10^−13^ → 7.08 × 10^−14^ (−33.33%)	N/A	58.49 → 57.42 (−1.83%)
[95]	Bentonitte	Chitosan, poplar hot water extract	39.3 → 52.9 (+34.69%)	11.7 → 7.9 (−32.28%)	N/A	8.71 × 10^−11^ → 6.72 × 10^−11^ (−22.85%)	3.36 × 10^−8^ → 1.63 × 10^−18^ (−51.48%)	280 nm: 35 → 28 (−20%)	N/A
[96]	Kaolinite	Chitosan	17.0 → 12.0 (−29.41%)	14.8 → 10.0 (−32.43%)	N/A	9.61 × 10^−11^ → 7.92 × 10^−11^ (−17.58%)	N/A	400 nm: 87 → 77 (−11.49%)	N/A
[97]	Sepiolite	Alginate, sepiolite-myrtle berries extract	38.0 → 117.0 (+207.89%)	3.8 → 3.2 (−15.79%)	2.0 → 4.3 (116.08%)	1.79 × 10^−6^ → 1.62 × 10^−6^ (−9.49%)	N/A	600 nm: 90 → 73.8 (−18.00%)	N/A
[47]	ZnO-doped MWCNTs	PVA	(+116.00%)	(+81.00%)	N/A	(−25.93%)	N/A	250 nm (−15.18%)	74.3 → 76.9 (+3.50%)
[98]	CNTs	Sodium alginate/chitosan	17.5 → 24.2 (+38.29%)	16.9 → 21.8 (+28.99%)	N/A	N/A	N/A	N/A	N/A
[99]	MWCNTs	GO, Ch.–PEO	N/A	N/A	N/A	N/A	N/A	400 nm: 76% → 21% (−72.37%)	N/A
[100]	Starch nanocrystals	Corn starch	17.4 → 20.3 (+16.67%)	16.7 → 10.6 (−36.52%)	N/A	3.58 × 10^−14^ → 2.89 × 10^−14^ (−19.27%)	N/A	800 nm: 74.8 → 54.0 (−27.80%)	N/A
[101]	Quinoa Starch nanocrystals	Cassava starch	6.5 → 16.5 (+153.85%)	10.2 → 7.1 (−30.39%)	2.8 → 6.6 (+135.71%)	1.5 × 10^−7^ → 1.2 × 10^−7^ (−16.67%)	N/A	N/A	15.0 → 23.1 (+54.00%)
[102]	CNFs	PVA	52.5 → 69.8 (+32.95%)	99.0 → 84.7 (−14.44%)	3578.0 → 4263.0 (+19.14%)	6.97 × 10^−7^ → 2.82 × 10^−7^ (−59.54%)	(−99.46%)	5.7 → 48.8	N/A
[103]	CNFs	Starch	8.9 → 16.5 (+85.39%)	83.2 → 9.0 (−89.18%)	289.0 → 743.0 (+157.09%)	12.0 × 10^−11^→ 8.65 × 10^−11^ (−27.91%)	N/A	380 nm: 72.2→ 40.0 (−44.60%)	N/A
[104]	CNCs	Starch	16.2 → 24.6 (+51.85%)	13.1 → 3.6 (−72.51%)	12.9 → 21.0 (+62.35%)	2.08 × 10^−10^ → 1.84 × 10^−10^ (−11.54%)	N/A	N/A	N/A
[105]	BC	Gelatin	3.2 → 1.1 (−65.63%)	102.07 → 186.04 (+82.27%)	50.1 → 0.9 (−98.30%)	3.06 × 10^−9^ → 2.47 × 10^−9^ (−19.28%)	N/A	N/A	N/A
[106]	CNP	Corn starch, thymol	7.7 →13.7 (+77.92%)	139.0 → 157.0 (+12.94%)	33.7 → 63.6 (+88.72%)	N/A	N/A	N/A	N/A
[107]	CNP	Starch	1.1 → 10.0 (+79.55%)	67.0 → 90.8 (+35.48%)	6.0 → 39.9 (+568.96%)	3.06 × 10^−15^ →1.75 × 10^−15^ (−42.81%)	(−48.12%)	N/A	N/A
[91]	SiO_2_ NPs	Chitosan	101.3 → 131.9 (+30.21%)	4.8 → 2.3 (−52.08%)	N/A	1.34 × 10^−11^ →1.06 × 10^−11^ (−20.90%)	N/A	400 nm: (−85.19%)	N/A
[108]	SiO_2_ NPs	PLA	43.3 → 34.7 (−19.86%)	2.6 → 2.9 (+11.54%)	1775.8 → 1840.6 (−3.65%)	3.81 × 10^−3^ →2.53 × 10^−3^ (−33.60%)	N/A	N/A	51.4 → 51.6 (−0.39%)

Notes: TS: tensile strength; EAB: elongation at break; WVP: water vapor permeability, OP: oxygen permeability, Tg: glass transition temperature (i.e., an amorphous polymer changes from the glassy state to the elastic or rubbery state), MMT: montmorillonite, PET: polyethylene terephthalate, PAA: polyacrylic acid, PVA: poly (vinyl alcohol), PLA: poly (vinyl alcohol), N/A: not applicable, CNTs: carbon nanotubes, MWCNTs: multi-walled carbon nanotubes; GO: graphene oxide, Ch.–PEO: chitosan–polyethylene oxide; CNCs: cellulose nanocrystals; CNFs: cellulose nanofibers, BC: bacterial cellulose, CNP: chitosan nanoparticles. +/−: increase or decrease in the parameters of the nanocomposite properties compared to controls. The transmittance of the film shows its UV (200–320 nm) and visible light (400–800 nm) barrier properties. The table includes control film values and optimal film values for each parameter.

## Data Availability

The original contributions presented in the study are included in the article, further inquiries can be directed to the corresponding authors.

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
