# Peer review of "Nanofillers in Novel Food Packaging Systems and Their Toxicity Issues"

_foods, 2024, doi:10.3390/foods13132014_

Round 1
Reviewer 1 Report
Comments and Suggestions for Authors
The manuscript entitled: ‘’Nanoparticles in Novel Food Packaging Systems and Their Toxicity Issues’’ provides a detailed characterization of nanoparticles in the food sector. The manuscript addresses the problems of smart packing and the toxicity of packing containing nanoparticles. Despite, was presented interesting topic I have a some comments.
Please update your literature review for 2021-2024 publications, because as the 2008-2011 publications do not represent the current state of knowledge. For example, in the subsection cellulose-base nanofibers or nanowhiskers, the authors can update the publications with the work entitled: ‘’Nano-/Microcapsules, Liposomes, and Micelles in Polysaccharide Carriers: Applications in Food Technology’’ (Appl. Sci. 2023, 13(21), 11610; https://doi.org/10.3390/app132111610). Similarly, the other chapters should be updated.
An important aspect that would enrich the work would be a table comparing the use of nanoparticles in packaging with the analysis of the content in food.
Why the authors characterize the same metal nanoparticles (their impact, toxicity, applications): Ag NPs, ZnO NPs, TiO2 NPs which are already characterized in the paper: ‘’Metal-Based Nanoparticles in Food Packaging and Coating Technologies: A Review’’ (Biomolecules 2023, 13(7), 1092; https://doi.org/10.3390/biom13071092), and they didn’t focus on enriching the work with those not described?
Chapter assessment methods for NPs is very poorly characterized.
Why did the authors not also describe zeolites, which are also used in the food sector for packaging, e.g. the paper entitled: ‘’Active Packaging of Button Mushrooms with Zeolite and Açai Extract as an Innovative Method of Extending Its Shelf Life’’ (Agriculture 2021, 11(7), 653; https://doi.org/10.3390/agriculture11070653).
I would suggest describing in more detail the interesting subsection that is the application of NPs to reduce mobility.
An additional attribute of the paper that would definitely increase the number of citations would be the listing of potential applicable films containing nanoparticles for example films containing encapsulated propolis in hyaluronic matrix (‘’Synthesis and Investigation of Physicochemical and Biological Properties of Films Containing Encapsulated Propolis in Hyaluronic Matrix’’; Polymers 2023, 15(5), 1271; https://doi.org/10.3390/polym15051271) as applicable/verifiable in the form of active packaging. The form of a table here would be the best way to present this information.
I also suggest enriching tables 1 with current work.
Comments on the Quality of English LanguageModerate editing of English language required
Reviewer 2 Report
Comments and Suggestions for Authors
Manuscript ID: foods-2803702
Comments:
This paper presents a review on use of various nano-fillers as modifiers for improving physical, functional and biological properties of developed food packaging materials and their possible cyto- and eco-toxicity. Some comments and suggestions for improving the quality of this article are as following:
- - For the title, “Nano-fillers” might be appropriated than “Nanoparticles”.
- - Line 62: “…. nanotechnology ….” should be replaced with “….. nano-fillers …”
- - Line 81: The term “Improved packaging” used for sub-heading of section 2 is not clear. Which properties were improved? It should be changed for example “Use of nano-fillers for physical properties improvement of food packaging”. Also, it can be used “Use of nanofillers in active food packaging” and “Use of nanofillers in intelligent food packaging” for the sub-heading of sections 3 and 4, respectively.
- - Line 232: “….. the density of cellulose nanofibers …..” should be replaced by “….. the concentration of cellulose nanofibers …..”
- - Line 236-254: the context provided are not relevant to the heading in section 2.4.
- - For section 2, the mechanisms and interactions for property improvements mediated by nanofillers should be intensively analyzed and summarized into proper schematic representations.
- - For section 4 (Intelligent food packaging), the literature review on intelligent abilities of packaging based on nano-fillers incorporation as active components in intelligent packaging should be further discussed.
- - Line 396-397: There was no this reference (Salarbashi et al., 2021) cited in the reference list.
- - Line 416-420: The provided context are not relevant to biodegradability.
- - Line 429: The results from the work of Rimdusit et al (2008) was not consistent with the study of Fukushima et al (2013). Please recheck.
- - Section 6.1: the sub-heading used (The Fick’s Law) is not appropriate for the context. The context reviewed mostly emphasized on the migration and some factors affecting the migration of nano-fillers from the materials. Additionally, the review on migration study of nano-fillers from packaging materials should be expanded in more details.
- - Section 6.2, lines 501-504 and 509-516, the contents provided are not relevant to the migration of nanofillers from packaging materials as for the safety issue.
- - Section 6.3 and 6.5 should be combined. They involved the toxicity concerns (cytotoxicity and ecotoxicity). The review relevant on studies showing the evidences for toxicity should be further and clearer presented and discussed.
Reviewer 3 Report
Comments and Suggestions for Authors
This review provides no new insights. Relevant new developments are neglected. The depiction of the state for the art is poor, no comparisons and high quality graphs for better depiction. Authors should derive high quality depictions with the values from literature, oxygen and water vapour barrier with and without NPs, depictions of equations describing the behaviour of NPs.
The main question is not answered: Why NPs in packaging are hardly used?
L. 12-13: That is discussed for decades, not “recently”.
L. 15: Only of “biodegradable polymers”?
L. 19-22: Too general. Be more specific.
L. 23-31: Too general.
L. 35-38: There are more: Information, containment, transport.
L. 42-43: Only polyester?
L. 44: 10 yr. is short. More references needed for this value.
L. 46: PET has low diffusion coefficients!
L. 53: For at least 3 decades!
L. 52: Look for the definition of biopolymer.
L. 66-68: Add values.
L. 70: These are often not approved.
L. 71-72: Use equations for description!
L. 74-76: There are more methods. Discriminate between solid and solved nanoparticle matter.
L. 82-86: Add values.
Table 1: Values and units for properties are missing.
L. 101: Most trials were disappointing!
L. 100-157: Too descriptive. Use equations, aspect ratio, provide table with values for polymers with and without NPs. Important in barrier log-depictions are relevant. Synthetic platelets are missing.
L. 144: Very low!
L. 158-177: Values are missing. How it compares to other methods such as orientation?
L. 178-207: Too descriptive. Provide table with values for polymers with and without NPs.
L. 209-254: Too descriptive. Provide table with values for polymers with and without NPs.
L. 256: That is in research for 50 yr. and used in small amount for 30 yrs.
L. 255-283: Too descriptive. Provide table with values for polymers with and without NPs.
L. 284-316: Too descriptive. Provide table with values for polymers with and without NPs. At Ag NPs the ions are active not the particles! Ag is immobilised by sulphur-compounds in foods.
L. 317-365: Too descriptive. Provide table with values for polymers with and without NPs.
L. 367-449: Too descriptive. Provide table with values for polymers with and without NPs.
Fig. 3: That are not NPs!
L. 451-497: Equations that describe the behaviour are missing!
L. 498-623: Too descriptive.
Reviewer 4 Report
Comments and Suggestions for Authors
Foods; foods-2803702
Dear authors,
thank you for the great effort represented in the manuscript “Nanoparticles in Novel Food Packaging Systems and Their Toxicity Issues”
Recommendation: Major revision.
- Novelty of the manuscript must be better emphasized.
- The abstract should be written more precisely without including unnecessary information. Try to highlight the novelty of the research and its contribution accurately.
3. The introduction section needs to be rewritten to incorporate recent work and emphasize the significance of using nanoparticles. The authors should provide a comprehensive overview of the field and explain how this particular nanoparticle for food packaging applications addresses current research gaps or challenges.
4. Please clarify the innovations of this work in the introduction section and compared to other work.
5. Rewrite the caption of Figure 3.
6. The authors must include the food packaging mechanism diagram.
7. The conclusion should be revised to reflect the larger findings of this manuscript.
- The authors should improve the food quality discussion using these recent references. https://doi.org/10.1002/pat.3847, https://doi.org/10.1016/j.envres.2023.116634.
- The authors can compare this research with other similar studies in a table format to highlight the advantages of their work.
- The text is not free from grammatical, format, and punctuation errors. Please ask a native English speaker to revise and proofread their revised manuscript before re-submission.
Ok
Round 2
Reviewer 1 Report
Comments and Suggestions for Authors
Manuscript with a changed title ‘’ Nanofillers in Novel Food Packaging Systems and Their Toxicity Issues’’ has been radically changed. The authors introduced changes in the abstract, keywords, or introduction. The authors added a graphical abstract.
The authors followed the recommended comments and suggestions.
The manuscript has been enriched with extensive tables entitled: ‘’The key features and findings of previous similar reviews’’, ‘’ Nanofillers are used in improved food packaging’’, ‘’ The application of nanofillers in biodegradable food packaging ‘’, ‘’The applications of nanofillers in intelligent packaging, i.e., to indicate the freshness 631 of food products through pH changes’’ or ‘’The antimicrobial properties of metal-based NPs in active packaging’’, that significantly increase the substantive value of the manuscript.
In addition, chapters entitled: ‘’Silver NPs (Ag NPs)’’ or ‘’ Use of nanofillers in active food packaging’’ have been refined in detail.
In addition, the authors developed and added important chapters: ‘’ Use of nanofillers in photodegradable and biodegradable food packaging’’, ‘’ Nanocomposites-based intelligent films’’, ‘’ EO-loaded Nanoemulsion ‘’or ‘’Other NPs’’ without which the work would not constitute a coherent whole.
Comments on the Quality of English Language
Minor editing of English language required
Author Response
We would like to express our sincere appreciation for your thoughtful and insightful comments on revised manuscripts “Nanofillers in Novel Food Packaging Systems and Their Toxicity Issues”. It was your suggestions that greatly enhanced the quality and depth of our work. We are very grateful for the valuable time you have taken to revisit and comment on the revised manuscripts.

Reviewer 3 Report
Comments and Suggestions for Authors
L. 19: The materials are not new.
L. 26-34: Be quantitative. This part is too general and too predictive. Present values.
L. 68: That is discussed for decades already.
L. 77-80: Too general. For drop-in solutions this sentence is wrong.
L. 86: What is “excellent”? Provide values.
L. 90: Which ones and why?
L. 97: Eq. 1, please look into the publications of “Bott” concerning dependence of diffusivity on molecular weight or possible size of the nanoparticle.
L. 113-114: Unclear.
Table 1: Belongs to Conclusions and should be written sentences.
L. 141: “silicon dioxide” NPs are used for decades already.
L. 161: What are typical L/D-values. These are too small for many applications.
L. 175-204: Fully missing, the barrier improvement was too low. How much was it? Important, barrier improvement should be > factor 10. These materials were developed 25 yrs. ago and it stopped because of poor barrier improvement compared to higher cost.
L. 175-384: Here no novelty. All what is written can be read in other papers in higher quality. To add value to these chapters authors could highlight developments in last 5 or 10 years and compare it with the status before. Verly likely is a negligible, unimportant development, what could be written.
Table 2: Make high quality graphs out of it, e.g., Elongation versus Youngs, WVP vs. oxygen permeability. Stability is useless without information about temperature and possible pre-drying.
L. 393-562: Nothing new. New could be how much ions are released and this could by compared with the concentration that is necessary.
Table 3: This Table is useless as it is. Best way is to compare the materials with and without NPs and provide values for that. Make a high quality graph out of the results.
L. 586-587: Where are these used?
L. 588:? Why are these NPs?
L. 586-633: pH-indicators can be solved. No need of NPs.
L. 634-700: No new. Work out new results.
Migration process: Compare diffusion coefficients in dependence of the molecular weight, see equations in work of “Bott”.
L. 887-909: Research on NPs in polymers in going on for 25 yrs. Why are these hardly be used? The conclusion do not provide new insights.
Reviewer 4 Report
Comments and Suggestions for Authors
The manuscript has been revised meticulously as per reviewers comments. Now, the MS may be considered for publication.
Comments on the Quality of English LanguageOk
Author Response
On behalf of my co-authors, we would like to thank you for reviewing this revised manuscript entitled “Nanofillers in Novel Food Packaging Systems and Their Toxicity Concerns” on your busy schedule. We are encouraged by the constructive comments and scientific suggestions from the editors and you on our revised manuscript, which have guided us step by step in revising the manuscript to its current state.
